# Host range and zoonotic potential linked to P-like fimbrial (PLF) adhesin specificity in avian pathogenic *Escherichia coli*

Fariba Akrami[1,2], Hossein Jamali[1,2], Sebastien Houle[1,2], Paula Armoa Ortiz[1], Charles Calmettes[1,2,3], Charles M. Dozois[1,2,3]*

**1** Institut National de la Recherche Scientifique (INRS), Centre INRS Armand-Frappier Santé Biotechnologie, Laval, Quebec, Canada, **2** Centre de Recherche en Infectiologie Porcine et Avicole (CRIPA), Faculté de Médecine Vétérinaire, Université de Montréal, Saint-Hyacinthe, Quebec, Canada, **3** Pasteur Network, Paris, France

* charles.dozois@inrs.ca

## Abstract

Fimbrial adhesins are surface-associated bacterial proteins that contribute to colonization, tissue tropism, and biofilm formation. Genomic analysis of the avian pathogenic *Escherichia coli* (APEC) strain QT598 identified a plasmid-localized fimbrial operon, termed *plf* which encodes P-like (PL) fimbriae. Herein, we investigated the role of P-like (PL) fimbriae, encoded on the ColV plasmid of APEC strain QT598, in a natural host turkey respiratory infection model. We determined that deletion of the *plf* genes reduced colonization in the lungs of turkeys. The PlfG class II fimbrial adhesin from APEC strain QT598 demonstrated species- and tissue-specific adherence, as adherence to turkey lung sections was more pronounced than adherence to chicken lung sections. *In vivo*, expression of *plf* was also found to be upregulated in the lungs of turkeys during infection, as determined by qRT-PCR. Glycan array analysis showed that the PlfG class II adhesin recognizes Lewis b, Lewis y, and H antigens as potential receptors that are present on human red blood cells and on the surface of turkey respiratory tissues, implicating specific α-1,2-linked glycans as receptors. Structural modelling of Plf adhesin revealed conserved β-sandwich folds with distinct binding pockets that predict receptor specificity differences between Plf adhesin variants and other fimbrial adhesins such as PapG II and the PlfG class I adhesin from UMEA 3703–1. These findings highlight the role of PL fimbriae in APEC virulence and suggest a potential zoonotic and foodborne risk from poultry to humans, demonstrating the common recognition of glycans present on both turkey and mammalian host cells and tissues.

**Data availability statement:** All relevant data are within the manuscript and its Supporting information files.

**Funding:** Funding for this work was supported by Natural Sciences and Engineering Research Council (NSERC) Canada Discovery Grant number RGPIN-2025-07045.NSERC funds were for a Discovery grant to C.M. Dozois. FRQNT funds for the multi-institutional Swine and Poultry Infectious Diseases Research Centre (CRIPA) grant. The funders had no role in study design, data collection and analysis, decision to publish, or preparation of the manuscript.

**Competing interests:** The authors have declared that no competing interest exist.

**Abbreviations:** ExPEC, Extra-intestinal *Escherichia coli*; APEC, Avian pathogenic *Escherichia coli*; UPEC, Uropathogenic *Escherichia coli*; UTI, Urinary tract infections; OMP, Outer membrane protein; LPS, Lipopolysaccharide; PAIs, Pathogenicity-associated islands; PLF, P like fimbria; WT, Wild type; STs, Multilocus sequence types; Col V, colicin V

## Author summary

Bacteria often use hair-like structures called fimbriae to attach to host cells and begin infection. In this study, we investigated a newly identified P-like (PL) fimbrial system in an avian pathogenic *Escherichia coli* strain isolated from a diseased turkey. By combining genetic, molecular, and infection-based approaches, we showed that PL fimbriae help the bacteria attach to lung tissue and colonize the turkey respiratory tract. Bacteria lacking these fimbriae were less able to colonize turkey lungs, and PL fimbriae promoted stronger attachment to turkey lung tissue than to chicken lung tissue. We also found that PL fimbriae recognize specific sugar molecules that are present on both turkey and mammalian cells. Together, these findings highlight how PL fimbriae contribute to APEC virulence and shape interactions between the bacterium and its hosts, while suggesting a possible relevance to cross-species exposure.

## Introduction

Avian pathogenic *Escherichia coli* (APEC) is one of the subsets of extra-intestinal pathogenic *E. coli* (ExPEC*),* and is responsible for multiple types of infections in poultry [1,2]. Colibacillosis can cause high mortality and morbidity in affected birds, including turkeys, ducks, and chickens [3,4]. There is also evidence that some *E. coli* from poultry may be a source of infections in humans [5–7]. Some ExPEC strains from either human or avian infections have similar genomes and contain genes that were horizontally transferred, which can contribute to the capacity of these strains to cause extra-intestinal disease in humans, poultry, and other animals [7,8]. Since ExPEC strains are considered pathobionts that can transiently colonize the intestinal tract of poultry and humans, transmission of certain ExPEC lineages from poultry to humans may represent a potential zoonotic risk via the food chain. Specific sequence types (STs) such as ST95, ST131, or serogroups like O1, O2, and O18 are isolated from poultry meat products and are closely related to some human ExPEC isolated from urinary tract infections (UTIs), sepsis, and neonatal meningitis. Further, strains belonging to serogroups O1, O2, and O78, as well as ST23 and ST95 are common types of APEC in poultry [1,9–11].

 *E. coli* strains exhibit remarkable genomic plasticity that facilitates the acquisition of virulence and antimicrobial resistance genes, enabling adaptation to hostile environments and colonization of extra-intestinal sites in both animal and human hosts [12]. These virulence factors are often located within conserved genomic regions known as pathogenicity islands (PAIs) or carried on large, conjugative plasmids, particularly those of the IncF incompatibility group. Among them, colicin V (ColV) plasmids are prominently associated with APEC and have also been identified in human-derived ExPEC strains, enabling them to adapt to unfavourable environmental conditions and enhance bacterial fitness and virulence [12,13].

In order to cause disease, APEC colonize host cells and tissues using a variety of adhesins such as fimbriae (pili) and curli [12,14]. Fimbriae are assembled at the cell surface through the chaperone-usher pathway and are classified based on morphology, specific receptor, and antigenic properties. One of the well-known types of fimbrial adhesins are the P-fimbriae or Pap-fimbriae, which were first identified from Uropathogenic *E. coli* (UPEC) and mediate adherence by attaching to glycolipid receptors on cells containing Gal-α-(1,4)-Gal galactoside residues [12,15–17]. Pap and closely related fimbriae have also been identified in other pathogenic *E. coli,* including *E. coli* causing systemic disease in swine, some APEC strains, and UPEC from dogs and cats [18,19].

The genome of an APEC O1 strain (QT598) isolated from a turkey with colibacilosis in 1995 was sequenced and found to harbor a ColV plasmid, containing a distinct region [19]. This region was shown to contribute to kidney colonization in a mouse UTI model and contains a novel member of the π-fimbrial family that we have called P-like (PL) fimbriae, encoded by the *plf* gene cluster [19]. The *plf* genes were identified on virulence plasmids in both UPEC and APEC. In a previous report first identifying and initially characterizing the *plf* gene cluster in 2022 [19], we identified 686 distinct entries in the NCBI database that contained *plf* sequences. These entries were sourced from 171 human infections (mainly urinary isolates or blood infections) and 66 human fecal isolates. As well, strains encoding *plf* sequences were associated with infections in dogs, and samples from turkeys, chickens, and other avian species, and a variety of environmental sources.

This new fimbria is a member of the π-fimbrial group that includes P fimbriae that are known to be important for ExPEC virulence. The two different PL fimbrial adhesin types identified in APEC QT598 (PlfG class II) and UPEC UMEA-3703–1 (PlfG class I) both agglutinate human and turkey erythrocytes, but they have distinct adhesin proteins that share only 56% amino acid identity (S9 Fig). The PL fimbriae from QT598 have been shown to contribute to competitive fitness in the kidney in a mouse urinary tract infection model, and mediate adherence to human bladder and kidney epithelial cells [19], however their potential role for systemic infection of poultry has not been investigated. Further, expression analysis has determined that during UTI in the mouse model, the *plf* genes are upregulated more than ten-fold compared to during culture *in vitro*. Taken together, these results suggest that PL fimbriae may recognize common or similar receptors on both human and avian cells/tissues and can contribute to systemic infection in both mammalian and avian hosts [19].

Herein, we investigated the role of the PL fimbriae for APEC strain QT598 in a turkey respiratory infection model and demonstrated its importance for colonization of specific tissues in young turkey poults, respectively. The receptor specificity of the PlfG class I and class II adhesins from UPEC and APEC strains were also investigated.

## Results

### Analysis of *plf* expression *in vitro* using a *plf* promoter *lux* fusion

Previous work indicated better expression of PL fimbriae following growth on LB agar plates compared to growth in LB broth [19]. To investigate the effects of serial passage on *plf* expression by strain QT598, we constructed a reporter strain in which the *plf* promoter region was cloned upstream of the *lux* operon. The strain (QT5828) was grown for three consecutive passages in LB with and without agitation, as well as on LB agar (LBA), at 37 °C. Bioluminescence from the reporter was measured after overnight growth following the first, second, and third passages for each condition. No significant differences in promoter activity were observed between passages in LB broth with or without shaking, and bioluminescence in LB liquid medium was low (S1 and S2 Figs). In contrast, *plf* expression was significantly higher during growth on LB agar compared to LB broth, with the highest activity detected during the third passage in late stationary phase.

### Generation of the *plf* mutant and complemented strains

To investigate the potential role of PL fimbriae during APEC infection in turkeys, we constructed a *plf* deletion mutant and a complemented strain, wherein the *plf* genes were re-introduced into the chromosome. Loss of PL fimbriae in the mutant and regain of expression in the complemented strain were confirmed by western blotting and hemagglutination tests.

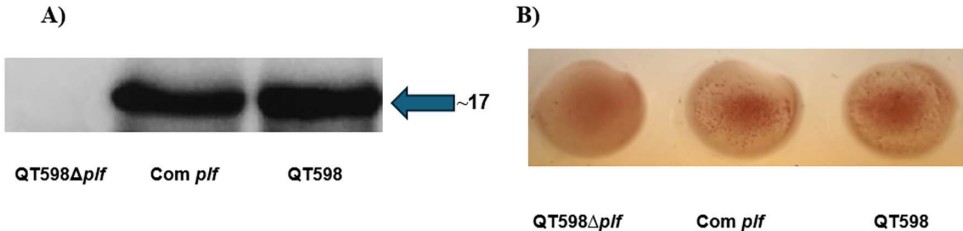

**Fig 1. Western blot and Mannose-resistant hemagglutination (MRHA) analysis of PL fimbriae expression.** A) Western blot using PlfA-specific antiserum. A PlfA-specific band was absent in the Δ*plf* mutant (QT4420), whereas a strong band was detected in wild-type strain QT598 and the complemented strain (QT6049). B) Hemagglutination with human erythrocytes in the presence of 2.5% D-mannose confirmed that MRHA was absent in the Δ*plf* mutant, whereas MRHA was present in wild-type parent QT598 and regained in the *plf* complemented mutant. Turkey erythrocytes gave similar results with no MRHA for the *plf* mutant, and positive MRHA for QT598 and *plf*-complemented strains (not shown).

Protein extracts were used to test for production of PL fimbriae using PlfA-specific antibodies by Western blot (Fig 1A). A strong band corresponding to the expected size of the PlfA subunit was detected in the wild-type and complemented strains but was absent in the Δ*plf* mutant (QT4420), indicating successful restoration of PL fimbriae production in the complemented strain (QT6049). Mannose-resistant hemagglutination (MRHA) of human or turkey erythrocytes in the presence of 2.5% D-mannose also confirmed the functional regain of MRHA in the complemented strain. By contrast, the *plf* mutant strain did not demonstrate any MRHA phenotype (Fig 1B).

### Deletion of *plf* results in decreased colonization in the lungs of turkeys

As strain QT598 was originally isolated from a 4-day-old turkey poult, we used air sac infection in turkeys as a natural host model for this APEC strain. Moreover, since PL fimbriae also mediate agglutination of turkey erythrocytes [19], we posited that these fimbriae might play a specific role in tissue colonization in the turkey. To evaluate the role of PL fimbriae in tissue-specific colonization of the avian respiratory tract, we developed an air sac inoculation model in 6-day-old turkey poults. Using this model, the wild-type QT598 strain was recovered in high numbers from the lungs of infected poults with colony-forming unit (CFU) counts ranging between ~$10^5$ to $10^7$ CFU/ml of tissue at 48 h post-infection (Fig 2A), indicating efficient colonization with high bacterial numbers in lung tissues. By contrast, the *plf* mutant (QT4420) had significantly less bacterial colonization in the lungs with bacterial burdens ranging from ~$10^2$ to $10^4$ CFU/ml. This represents a 2–3 $\log_{10}$ decrease (Fig 2A) when compared to the wild-type parental strain QT598 (P < 0.0001). Importantly, when the *plf* mutant was complemented by re-introduction of *plf* genes, the complemented strain (QT6049) regained a capacity to colonize the lungs that was significantly higher than the *plf* mutant (P < 0.01) and bacterial burden in the lungs was not significantly different to the wild-type parent (Fig 2A). These results demonstrate that PL fimbriae play an important role in promoting bacterial colonization of turkey lung tissue.

To determine whether PL fimbriae also influence bacterial presence in extra-pulmonary tissues, bacterial burdens in the liver and spleen were assessed 48 h post-infection. The spleens and livers of 5/10 poults infected with wild-type QT598 had ~$10^3$ CFU/ml, whereas no bacteria were recovered from the other poults infected with QT598. For the *plf* mutant, only 2/10 and 1/10 poults had ~$10^3$ CFU/ml in the liver and spleen, respectively, indicating a decreased capacity for the *plf* mutant to colonize these tissues (Figs 2B and 2C). For the complemented mutant, 3/10 and 4/10 poults had ~$10^3$ CFU/ml in the liver and spleen, respectively. Bacteremia was also assessed 48 hours after infection, but no bacteria were detected in the blood from any of the samples. These findings indicate that while QT598 efficiently colonizes the respiratory tract, bacterial recovery from extra-intestinal organs is limited and does not reflect sustained bacteremia.

Remarkably, infected poults showed little to no clinical signs of colibacillosis, such as respiratory distress, lethargy, or huddling, even in the presence of significant pulmonary bacterial loads. However, a difference in percentage weight gain

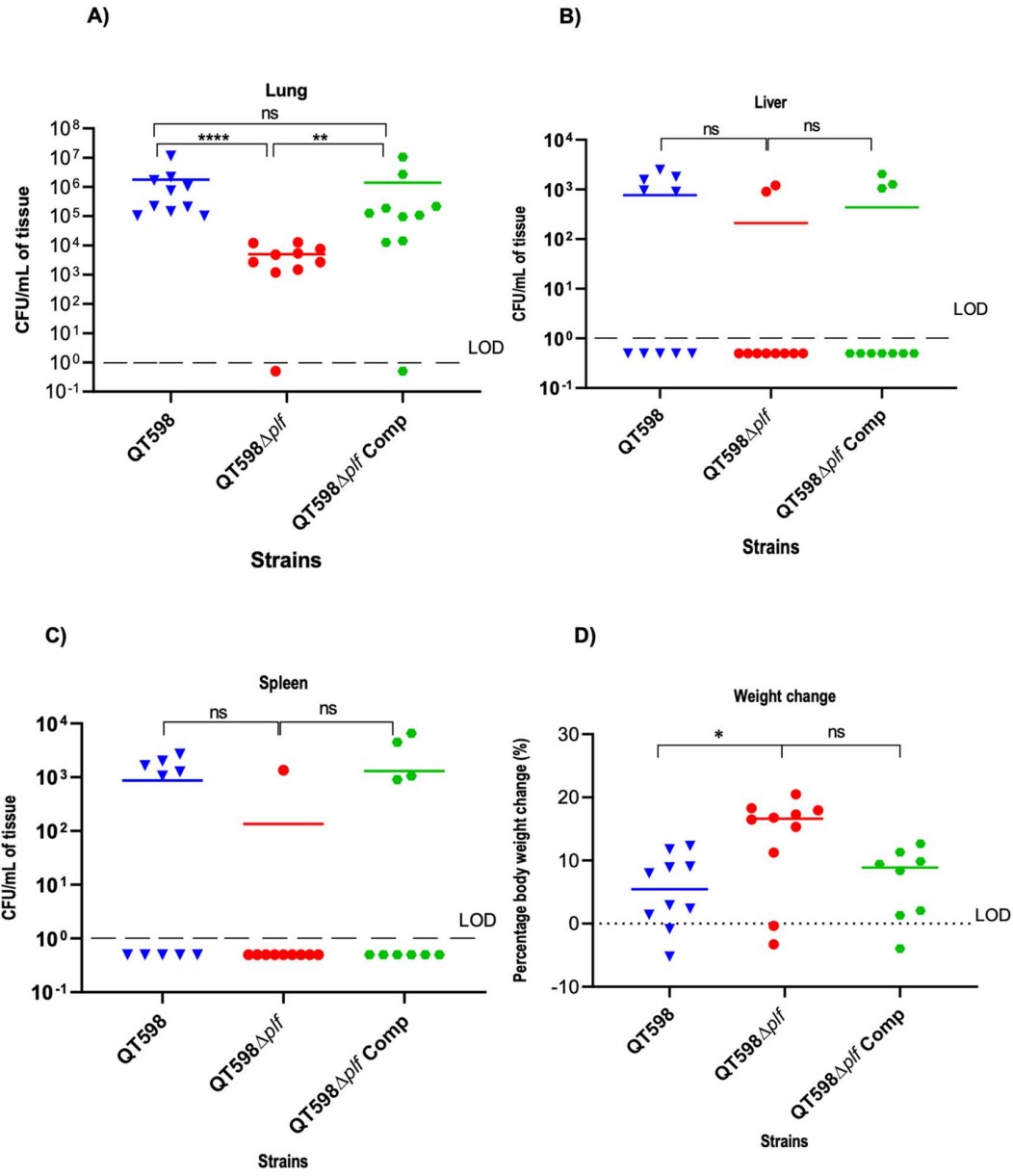

**Fig 2. Bacterial colonization levels in turkey organs following infection with QT598, QT598Δ*plf*, and the complemented strain.** The 6-day-old turkeys were infected, and bacterial viable counts were obtained from the right lung (A), liver (B), and spleen (C) at 48 hours post-infection (n = 10 poults per group). Each symbol represents one sample from an individual poult, and horizontal bars indicate median CFU/ml of bacteria. The limit of detection (LOD) is indicated by the dashed line. For normally distributed data, one-way ANOVA with Tukey's post-hoc test was applied; for non-normal data, Kruskal-Wallis with Dunn's post-hoc test was used. In the lungs, deletion of *plf* significantly reduced colonization compared to the wild-type strain (****, $P < 0.0001$), while complementation partially restored colonization in the lung ($p < 0.01$). No significant differences (ns) were observed in the liver or spleen between groups. (D) Percentage of weight change in poults 48h post-infection. *, $P < 0.05$; **, $P < 0.01$; and ***, $P < 0.001$, by one-way ANOVA.

was observed among groups infected with the *plf* mutant compared to the wild-type or complemented strains. Deletion of the *plf* genes significantly increased the percentage of body weight in turkeys compared to the wild-type QT598 strain ($p < 0.05$). However, complementation of the *plf* deletion strain reduced the percentage weight change to levels similar to

the wild-type (Fig 2D). No significant weight difference was found between the Δ*plf* and Com *plf* groups. These results suggest that PL fimbriae contribute to subclinical effects on host fitness, likely associated with increased bacterial colonization of the respiratory tract.

Together, these data support a role for PL fimbriae in tissue-specific colonization of the avian respiratory tract and highlight the turkey air sac model as a relevant system for examining host–pathogen interactions during APEC infection.

### The expression of *plf* is increased in turkey respiratory tissues during infection

We investigated whether the expression of the *plfA* gene, encoding the major fimbrial subunit protein, varied under *in vitro* growth conditions or during infection of the turkey respiratory tract. The level of expression of *plfA* by strain QT598 *in vivo* was compared to growth on LB agar plates after 3 passages. Each sample was analyzed in three technical replicates across three independent experiments, and expression was normalized to the housekeeping gene *rpoD*. During infection, *plfA* was more expressed in the lung than in the air sac. When compared to *in vitro* growth, *plfA* expression was markedly elevated in both respiratory tissues (Fig 3). The air sac showed a ~2.6-fold increase ($P < 0.01$) in transcript abundance compared to *in vitro* growth, whereas expression in the lung was ~6.8-fold higher compared to expression levels *in vitro*. According to these findings, although production of PL fimbriae *in vitro* is favored after 3 passages on LB plates, levels of *plf* exression were even more highly expressed in turkey lung tissues and air sacs. Collectively, these findings indicate that *plfA* expression is not uniform across respiratory tissues but instead reflects a targeted adaptation to the air sac and lung microenvironments, potentially contributing to the efficiency of colonization and the overall virulence potential of strain QT598 in turkeys.

### PL fimbriae mediate adherence to turkey lung sections compared to chicken lung sections

To determine whether PL fimbriae contribute to host- and tissue-specific adherence, we used confocal microscopy to compare bacterial attachment to turkey and chicken lung sections. Confocal microscopy clearly demonstrated a difference in bacterial adherence between turkey and chicken lung sections. In turkey lung sections (Figs 4A, B, E, and F), GFP-expressing bacteria (green) were abundant and closely associated with the WGA-labeled epithelial membranes (red).

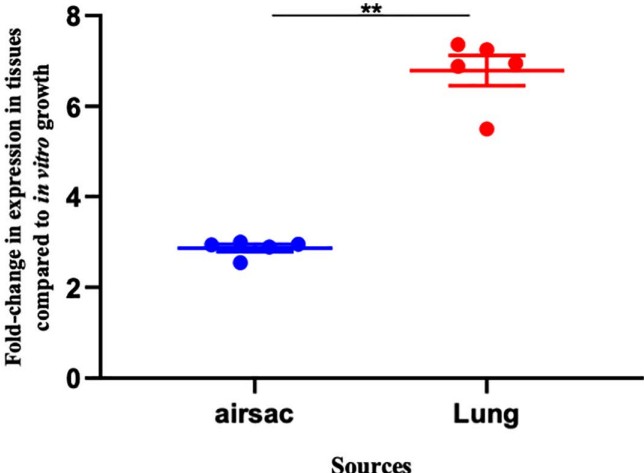

**Fig 3. Differential expression of *plfA* in the air sac and lung of infected turkey poults compared to LBA.** Quantitative RT-PCR analysis showing the fold change in *plfA* expression in bacteria recovered from the air sac and lung tissues, compared to bacteria grown on LB plates. Expression of *plfA* was significantly higher in the lung compared to the air sac (**, p < 0.01). Data represent the mean ± standard errors of the mean (SEM) from biological replicates.

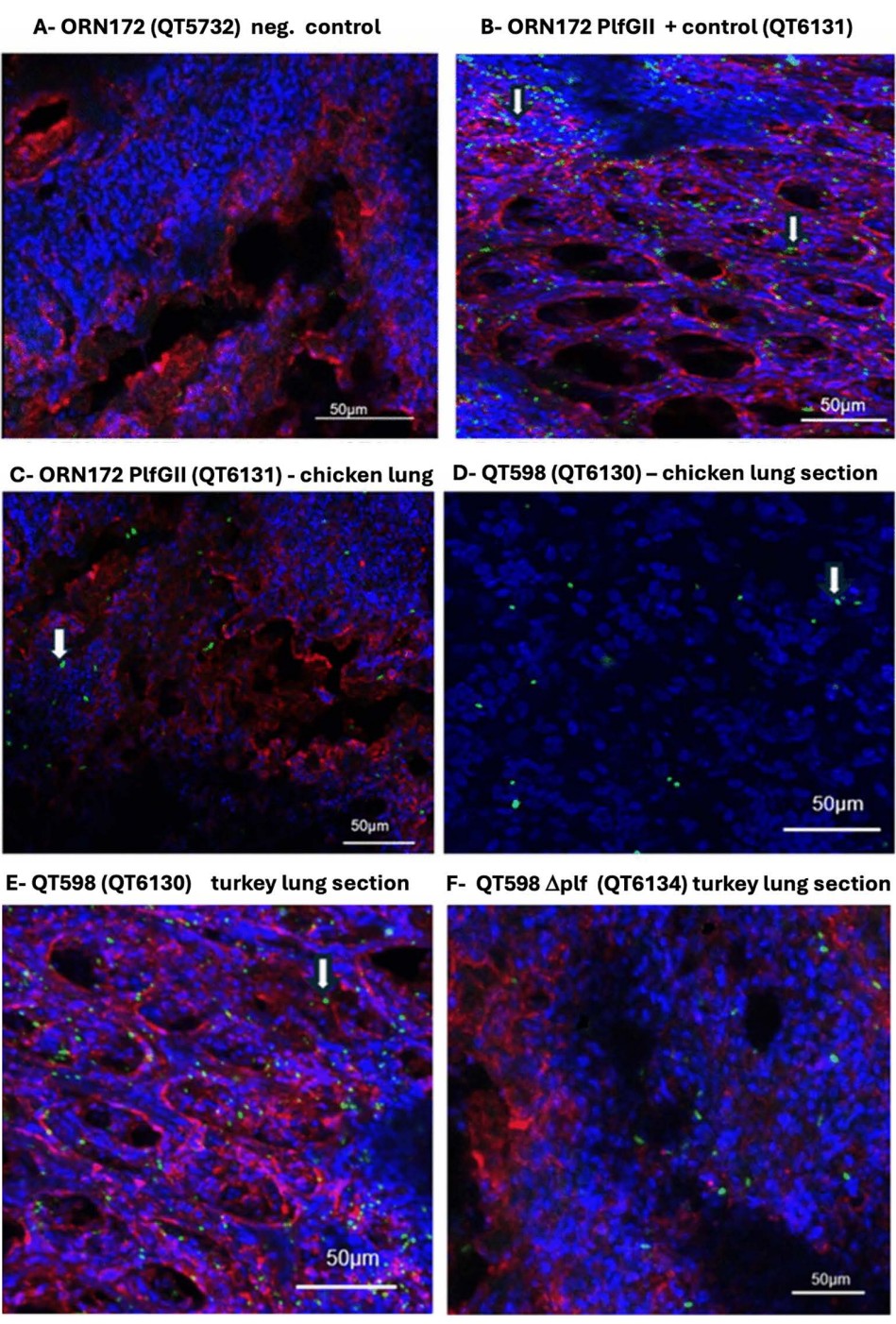

**Fig 4. Binding of GFP-expressing bacterial strains to OCT-embedded lung tissue from six-day-old turkey poults and chicks.** Confocal micros-copy revealed strong bacterial attachment to turkey lung epithelium, while minimal binding was observed in chicken lung tissue. (A) Lung tissue section from turkey, negative control ORN172 (QT5732). (B) Lung tissue section from turkey, clone expressing PlfG II (QT6131). (C) Lung tissue section from chicken, clone expressing PlfG II (QT6131). (D) Lung tissue section from chicken, tested with strain QT598. (E) Lung section of turkey tested with QT598. (F) Lung section of turkey tested with QT598Δplf. Nuclei of cells were stained with DAPI (blue), epithelial membranes with WGA-Alexa Fluor 647 (red), and bacteria expressed GFP (green). Images are representative of three independent experiments. Z-stack images were acquired using a Zeiss LSM 780 confocal microscope, and intensity calibration was performed using ZEN Black software. The white arrows show examples of green colored bacteria in each field when visible.

Many bacteria formed clusters or microcolonies along the epithelial surface, particularly surrounding the openings of air capillaries. The DAPI-stained nuclei (blue) revealed the well-organized nuclei of the turkey lung cells, with bacteria distributed extensively across the epithelial boundary. However, the Δplf mutant showed less attachment in comparison with WT strain QT598 to the turkey's lung sections (Fig 4F). In contrast, bacterial adherence to chicken lung sections (Fig 4C and D) displayed only a sparse GFP signal, with a smaller number of bacteria scattered as individual cells and lacking the dense clustering seen with adherence to turkey lung tissue. The epithelial membranes and nuclei were clearly visible, but bacterial association was minimal. Semi-quantitative analysis of confocal images revealed a significantly higher number of GFP-positive bacteria on turkey lung sections compared to chicken lung sections (approximately 50–60 vs 10–15 bacteria per field) by ImageJ software. All the pictures were magnified 100x and the experiments were repetaed 3 times. These observations indicate that strain QT598 grown under conditions that favor expression of PL fimbriae exhibit a species-specific tissue tropism for attachment to turkey respiratory epithelium compared to chicken respiratory epithelium that suggest that there are specific receptors on turkey lung tissues that are recognized by the PL fimbriae.

## Hemagglutination (HA) and adherence by PL fimbriae are inhibited by L-fucose and D-galactose

Macro-hemagglutination tests performed in the presence of 2.5% D-mannose (to inhibit type 1 fimbriae) revealed strong binding of PL fimbriae–producing strains (both positive control ORN172 expressing PlfG class II cloned from strain QT598 and the wild-type strain QT598) to both turkey and human O$^+$ erythrocytes, whereas the QT598Δplf mutant showed no hemagglutination (Fig 4). Sodium metaperiodate treatment is used to demonstrate that hemagglutination is carbohydrate-dependent, as this chemical oxidizes and eliminates carbohydrate from cell surfaces. Furthermore, QT4741 (PlfG class I adhesin cloned from strain UMEA-3703–1) also strongly agglutinated human erythrocytes, although this agglutination was not inhibited by sodium metaperiodate. On the other hand, sodium metaperiodate treatment of erythrocytes abolished hemagglutination in the positive control (ORN172 expressing PlfG class II cloned from strain QT598) and the wild-type QT598 strain (S3 and S4 Figs), further supporting that the PlfG class II adhesin of strain QT598 acts as a lectin that binds to specific carbohydrates. This suggests that binding by the PlfG class I adhesin from UMEA-3703–1 is distinct from the PlfG class II adhesin and may also include receptors other than carbohydrates at the cell surface. Addition of L-fucose or D-galactose completely inhibited HA by strain QT598, indicating that P fimbrial binding is mediated, at least in part, by recognition of specific carbohydrate moieties on the erythrocyte surface (Fig 5A). By contrast other sugars, including glucose, maltose, mannose, D-fructose, L-rhamnose, D-xylose, trehalose, sucrose, melibiose, maltose, N-Acetyl-D-glucosamine, N-Acetyl-D-galactosamine, and N-acetyllactosamine, as well as neuraminidase treatment, had no effect on hemagglutination. Since L-fucose and D-galactose were both found to inhibit HA (Figs 5A and S5A), inhibition by these sugars was tested at different concentrations. HA inhibition assays in 96 well-plates to determine the minimal HA inhibition concentration for L-fucose and D-galactose were 15 mM at titers as low as 1:64 with turkey and human O$^+$ erythrocytes (S5B Fig). No hemagglutination was observed above this dilution, indicating complete inhibition at the tested concentrations. Collectively, these results suggest that PL fimbriae preferentially recognize fucosylated and galactosylated glycoconjugates present on host cells.

To assess if PL Fimbrial binding to turkey lung sections could also be inhibited by specific sugars, the bacterial adherence assay was repeated in the presence of D-galactose or L-fucose at a final concentration of 50 mM. This concentration was used because lung tissues are a more complex binding matrix than red blood cells. The PL fimbriae attachment to the turkey sections was markedly decreased in the presence of D-Galactose and L-Fucose, which indicated carbohydrate-specific binding (Fig 5B). Presence of either of these sugars resulted in a 90–95% reduction in bacterial cells adhering to the turkey lung sections.

## PL fimbriae mediate adherence to human kidney epithelial cells

In a previous report, it was determined that PL fimbriae, which can be present in some avian pathogenic as well as uro-pathogenic *E. coli* from human infections could mediate adherence to urinary tract epithelial cells [19]. In adherence tests

A) Hemagglutination inhibition

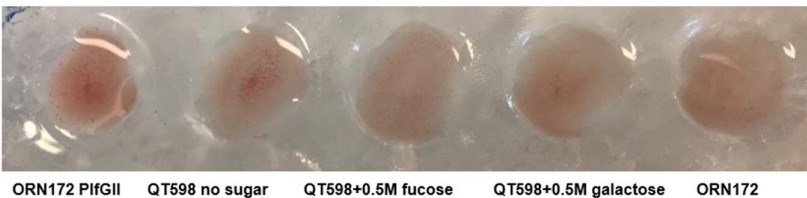

ORN172 PlfGII    QT598 no sugar    QT598+0.5M fucose    QT598+0.5M galactose    ORN172

B) Adherence inhibition

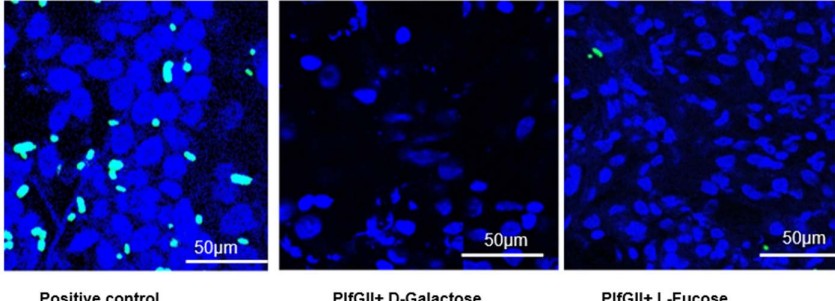

Positive control       PlfGII+ D-Galactose       PlfGII+ L-Fucose

**Fig 5. Inhibition of hemagglutination (HAI) of human O⁺ erythrocytes and adherence to turkey tissues by L-fucose and D-galactose.** A) Positive control is HA with strain QT5726 (Clone PlfG class II from strain QT598), QT598 HA with no sugar. L-fucose and D-galactose were shown to inhibit HA by strain QT598, and the negative control is QRN172. Agglutination inhibition was visualized after 30 min of incubation on ice. B) Confocal microscopy of DAPI-stained nuclei (blue) in turkey lung tissue sections after incubation with GFP-labelled bacteria in the presence of 50 mM D-Galactose or L-fucose. From left to right: The left is the positive control (QT6131) without any sugars. Then the middle and last ones are QT6131 after incubation with D-galactose or L-fucose, which showed that bacterial adherence to the lung sections was markedly reduced in the presence of either of these sugars.

with the HK293 human kidney epithelial cells, clones or strains producing PL fimbriae demonstrated significantly higher binding in comparison with those not expressing these fimbriae. Strains QT5726 (ORN172 expressing PlfG class II cloned from strain QT598), QT598, QT4741 (PlfG class I adhesin cloned from strain UMEA-3703–1) and UMEA 3703–1 exhibited the highest adherence at about $10^7$ CFU/ml (Fig 6). For both strains QT598 and UMEA 3703–1, deletion of *plf* genes led to a marked reduction in bacterial adherence to the kidney cell lines with a decrease of approximately 2–3 $\log_{10}$ units compared to their parental strains (P < 0.0001) which confirmed the role of PL fimbriae for bacterial adherence to these cells.

To further investigate the role of carbohydrate recognition in adherence, the cell line assay was repeated in the presence of inhibitor sugars (D-galactose and L-fucose) at a concentration of 50 mM (S6 Fig). The addition of these sugars for clones expressing PL fimbriae from QT598 (PlfG class II) and WT strain QT598 caused a sharp reduction in bacterial attachment (****P < 0.0001), suggesting they may be present within specific host cell receptors.

### Glycan array analysis indicates PlfGII adhesin recognizes Fuc-α-1,2-Gal containing complexes

A glycan array containing various oligosaccharide structures was incubated with dye-labeled strains. After washing to remove unbound bacteria, distinct signals indicated that the bacteria bound to specific oligosaccharides or polysaccharides spotted onto the arrays across all four chambers on a slide, and this binding was visualized via confocal microscopy. The spot distribution was provided by Ray Biotech (S7A Fig). We used a RayBio Glycan Array (S7B Fig). Under confocal microscopy, specific binding patterns were seen after probing with biotin-labeled lectin mixes (Fig 7A). Strong positive controls at the array corners verified the validity of the signals. Assay reliability was confirmed by the positive control strain QT5230 constitutively expressing type 1 fimbriae (*fimS* switch locked-ON), which demonstrated precise detection of the

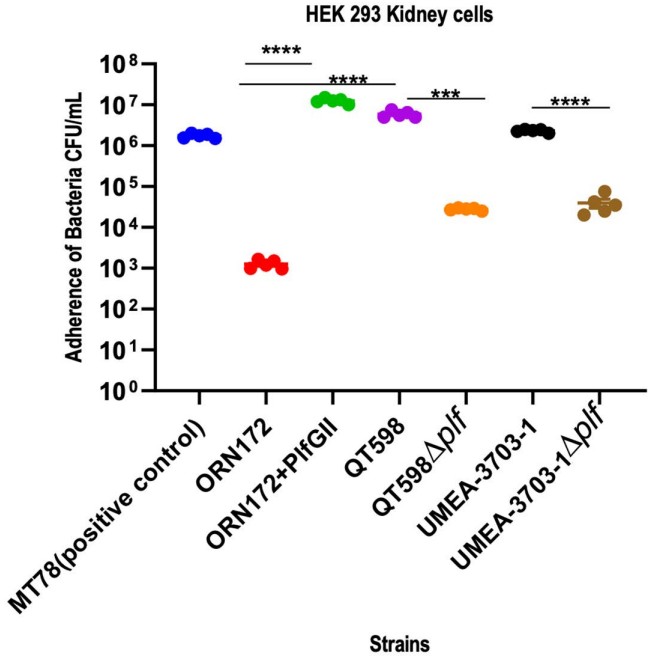

**Fig 6. Adherence of strains to HEK 293 kidney cells.** Monolayers were infected for 2 h, and adherent bacteria were quantified by plating on LB Agar. Data are expressed as CFU/mL; bars represent means ± SEM from three independent experiments. ****P < 0.0001; ***P < 0.001; ns, not significant.

mannose spots (α-Man-Sp) on the array (Fig 7B). The negative control strain (ORN172) did not bind to any spots. Most signals from QT5726 (ORN172 PlfG Class II) clustered at positions corresponding to Lewis b (Le^b) ((Fuc-α-1,2)-Gal-β-1,3-(Fuc-α-1,4)-GlcNAc-β- [Lewis B]-Sp1), Lewis y (Le^y) ((Fuc-α-1,2)-Gal-β-1,4-(Fuc-α-1,3)-GlcNAc-β- [Lewis Y]-Sp1), and H antigen (Fuc-α-1,2-Gal-β-1,4-GlcNAc-β- [Blood H antigen trisaccharide]-Sp1) which are all (Fuc-α-1,2)-Gal containing structures (Fig 7C and D). Each of the sugars is replicated four times per chamber, and QT5726 (ORN172 expressing PlfG class II cloned from strain QT598), expressing the PlfG class II fimbrial adhesin, bound to Lewis Y, Lewis B, and H antigens at three of four spots. By contrast, spots for unrelated saccharides—including sialylated structures, mannose-rich glycans, and galactose-only motifs—showed minimal background fluorescence, indicating limited or no binding. Unlike the clone expressing the PlfG class II adhesin, the clone expressing PlfG I did not bind any saccharides on the slide. These results demonstrate that, despite some sequence similarity between QT598 and UMEA3703–1 PL fimbriae [19], these two PlfG adhesins recognize entirely different receptors. In the case of the PlfG class II adhesin, the molecules identified are known to be expressed on both human erythrocytes and the avian respiratory epithelium of some species, offering a potential molecular explanation for the hemagglutination of both O⁺ human and turkey erythrocytes, as well as promotion of adherence to human kidney epithelial cells and colonization of turkey lungs mediated by PL fimbriae.

To confirm specificity of binding to human erythrocytes, hemagglutination inhibition was performed with purified Lewis B and H antigens. The strain QT5726 (ORN172 expressing PlfG class II cloned from strain QT598) showed no hemagglutination after addition of either of these compounds (S8 Fig).

## Structural modeling

The PlfG predicted models, generated with AlphaFold and compared to validated structure using PyMOL, share a conserved β-sandwich fold, similar to the lectin domains of the well characterized FimH and PapG-II adhesins, which mediate glycan recognition [20,21]. The overall shape of the sugar-binding pocket is similar between different adhesins, suggesting

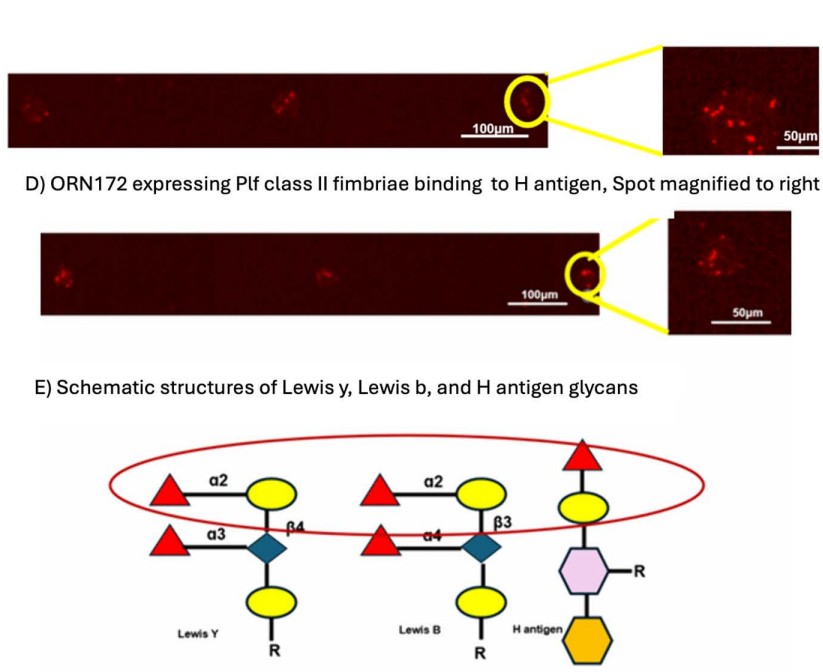

**Fig 7. Confocal microscopy of glycan arrays showing adherence specificity of PL fimbriae to (Fuc-α-1,2)-Gal containing glycans.** A) This is one chamber of the RayBio Glycan Array 100, which shows the binding patterns. B) QT5230 (*fim* locked-on strain) tested to promote attachment to the a-mannose spots as an experimental control, C) and D) show bacterial attachment of QT5726 (ORN172 expressing PlfG class II fimbriae) to the Lewis B and H antigens. QT5726 bound to 3 spots out of 4 replicates (indicated with yellow circle) to Lewis B, Lewis Y and H antigen (Lewis results not shown). E) The structure of Lewis B, Lewis Y, and H antigens demonstrating that all three structures contain a common (Fucose-α-1,2)-Galactose residue predicted to be the binding site for the PlfG class II adhesin.

a conserved mechanism of sugar recognition (Fig 8A and 8C). Although the overall folds are conserved, the predicted binding pockets differ in size, shape, and electrostatic properties, likely reflecting adaptations for distinct glycan specificities. These differences illustrate why FimH recognizes mannose, whereas PapG-II binds Gal-α1–4-Gal, and PlfG recognizes fucose-α1–4-Gal moieties. In PlfG class II (predicted with AlphaFold), the surface pocket residues maintain the

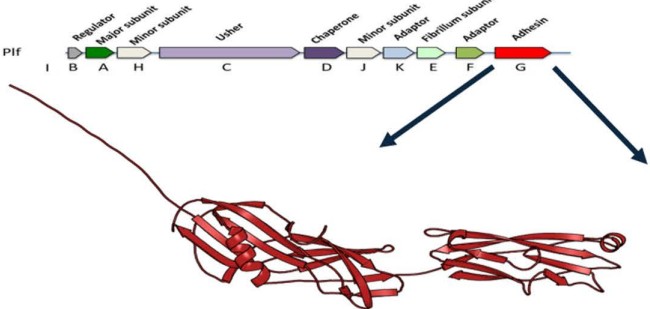

A) PlfG predicted structure in QT598

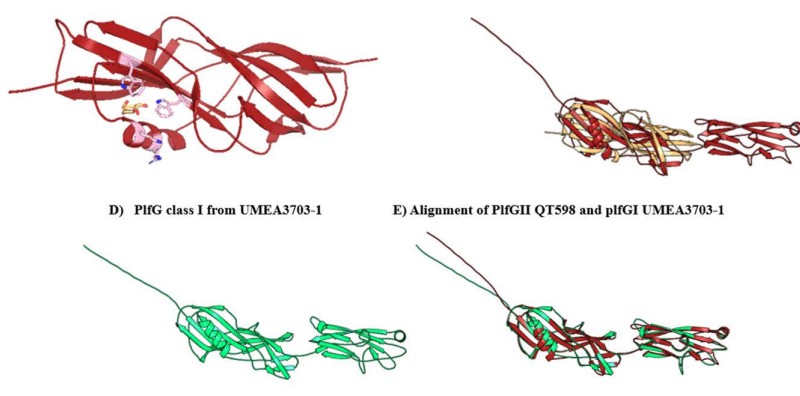

B) Predicted sugar-binding sites in PlfG of QT598     C) Alignment of PlfG of QT598 and PapGII

D) PlfG class I from UMEA3703-1     E) Alignment of PlfGII QT598 and plfGI UMEA3703-1

F) Predicted alignment with binding residues of PlfGII QT598 (red) and PlfG I UMEA3703-1 (green).

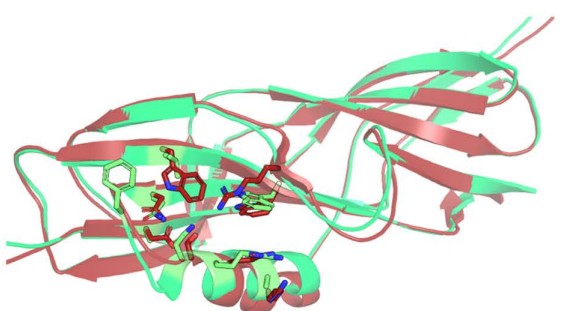

**Fig 8. Predicted 3D structure of the PlfG adhesins from QT598 (class II) and Umea 3703-1 (class I) modelled using AlphaFold.** A) The PlfG class II protein exhibits a classical β-sandwich fold typical of fimbrial tip adhesins. B) Residues potentially involved in glycan binding for galactose and fucose are shown as sticks, and selected residues predicted to be involved in sugar recognition are labelled (pink). C) Structural superposition of PlfG II (red) with PapG-II fimbrial adhesin from UPEC (yellow, PDB: 4Z3E). The core β-sandwich fold is conserved, while structural variation is observed in loop regions near the sugar-binding sites. The PlfG binding pocket appears narrower, consistent with its affinity for fucosylated Lewis B, Lewis Y and H antigens. The co-crystallize glycan of PapG II is shown as spheres to highlight the glycan binding site. D) The PlfG class I adhesin of UMEA 3703-1 exhibits a classical β-sandwich fold typical of fimbrial tip adhesins. E) Alignment of PlfG class II and class I structures which shows similarities although primary amino acid sequences differ considerably (They share 56% identity; see S9 Fig). F) Comparison of predicted sugar-binding residues in the putative glycan-binding sites of PlfG class I and II from UMEA3703-1 and QT598, respectively.

hydrophilic and aromatic character typical of sugar-binding sites in FimH and PapG-II, yet PlfG II displays distinct structural difference within the adhesin domain (Fig 8C). Accordingly, the predicted Lewis antigens-binding site of PlfG involves residues mediating Galactose and Fucose recognition, including tryptophan, arginine, and glutamine residues, which are highlighted in Fig 8B. Fig 8A and D show the models for PlfG of QT598 and UMEA 3703–1 which have similar predicted structures (Fig 8E) even though their PlfG primary amino sequences are considerably different (S9 Fig). However, Fig 8F shows the comparison of predicted binding sites of PlfG from QT598 and UMEA3703–1, illustrating the distinct side-chain composition within their glycan-binding pockets.

## Discussion

Fimbrial adhesins contribute to bacterial colonization of host tissues and can also promote biofilm formation [22]. One of the well-known chaperone-usher fimbriae of *E. coli* are P fimbriae that facilitate tissue-specific adherence of uropathogenic *E. coli* associated with UTIs [23]. Fimbriae can be encoded on either the bacterial chromosome or on plasmids in some *E. coli* strains [12]. In the current study, we have investigated the role of PL fimbriae produced by strain QT598, encoded by the virulence plasmid pEC598, in a turkey air sac infection model and also determine aspects of receptor specificity of this fimbrial adhesin.

Previously, PL fimbriae from strain QT598 were shown to provide a competitive advantage for colonization of the kidney in a mouse UTI model, and these fimbriae mediated adherence to both human bladder and kidney epithelial cells [19]. Since this strain was isolated from an infected turkey, we aimed to investigate whether these fimbriae contribute to respiratory tract colonization in a natural host model. Preliminary tests in the chicken indicated that strain QT598 was not effective at systemic infection in the three-week-old chicken model [24]. This could possibly be due to genomic differences compared to other APEC or ExPEC strains that may be able to cause disease in multiple host species, and that some APEC strains may have a more host-species-specific tropism [25]. This may also be due to functional differences between the chicken and turkey immune defenses and physiology [26]. This also suggested that certain APEC strains may be more adapted or able to infect the turkey host. Since APEC QT598 was initially isolated from a young turkey poult, herein we investigated the potential role of this APEC strain in a turkey respiratory infection model.

Deletion of the *plf* genes encoding PL fimbriae from strain QT598 resulted in a significantly reduced bacterial burden in the lungs of infected turkeys. Further, the reintroduction of *plf* genes complemented the infective capacity of the *plf* mutant, fulfilling molecular Koch's postulates regarding a collective role for these genes for systemic infection in the turkey model. This indicates that, similar to P fimbriae present in some APEC strains, PL fimbriae can mediate lectin-specific adherence to target tissues in the avian respiratory tract [27]. Notably, although systemic dissemination to the spleen and liver occurred at minimal levels in poults, no bacteria were identified in the blood, suggesting that PL fimbriae predominantly promote local tissue colonization rather than systemic bacteremia. These data align with prior publications emphasizing the significance of fimbrial adhesins in the host- and tissue-specific colonization by APEC strains [28,29]. The minimal effect on clinical signs, together with differences in percentage weight gain, indicates that PL fimbriae-mediated colonization may subtly influence host physiology and subclinical fitness [30–32]. Increased expression of *plfA* during infection in the turkey lung, compared to expression following *in vitro* growth on LB agar, suggests that environmental or host-derived signals within the lung microenvironment may upregulate production of PL fimbriae in turkey respiratory tissues. In line with this result, the expression of P fimbriae, which is controlled by phase variation under the control of environmental signals, can promote UPEC to colonize various host niches. So, the upregulation of PL fimbriae in the turkey lungs parallels colonization and increased expression of P fimbriae by APEC in the lower respiratory tract of chickens during respiratory infection [12,29]. Based on the adherence assays as well as glycan array tests, it was determined that PL fimbriae recognize sugars containing a Fuc-α-1,2-Gal structure, as the adherence and hemagglutination were inhibited by either fucose or galactose. The unique feature of these sugars is the presence of galactose and fucose residues [33], which we have identified as inhibitors of hemagglutination of turkey and human O+ blood group red blood cells.

The O+ blood type is characterized by the expression of the H antigen (Fucα1–2Gal), which serves as the precursor to A and B antigens [34]. The H antigen attaches to O+red blood cells but cannot bind to O- blood cells, providing evidence that PLF specifically agglutinates only O+RBCs. This suggests that PlfG class II may be a lectin-like binding protein that recognizes H antigen or fucosylated galactose terminal moieties. Additionally, Lewis antigens are fucosylated, secreted into mucus, or expressed on epithelial cells, depending on the activity of FUT2/FUT3 enzymes that modify lactosamine backbones [28,33,35]. It has been reported that avian mucins in the respiratory tract contain fucose, galactose, and complex N-glycans, including fucosylated structures [35, 36]. These could serve as potential sites for bacterial colonization; however, there is limited evidence regarding the lung mucosa of turkey [36]. It has been reported that chickens have reduced levels of Lewis antigens in their lungs [36], which could explain the minimal attachment of PlfG class II-mediated adherence to chicken lung sections. Interestingly, potential receptors for the PL fimbrial adhesin from strain QT598 were likely more present on turkey lung tissues and erythrocytes, since PL fimbriae mediated much stronger adherence to turkey lung and hemagglutination of turkey erythrocytes. By contrast adherence to chicken lungs or agglutination of chicken erythrocytes by PL fimbriae was minimal. This may partly explain why this strain appears to be more adapted to respiratory infection in turkeys than in chickens. Since the recognized sugars include structures present on human epithelial cells such as Lewis Y, Lewis B, and H antigens, PL fimbriae may potentially mediate adherence to human cells, suggesting a possible basis for cross-species colonization.

Interestingly, Lewis antigens and H antigen have been identified as important receptors for other pathogens including *Helicobacter pylori* and some noroviruses. Specifically, the *H. pylori* BabA adhesin mediates adherence to Lewis Y antigen, and this can promote disease through binding to the gastric epithelium [37–39]. Genogroup I norovirus strains also recognize Lewis B antigens on cells in the intestine, and this can result in tissue tropism of the virus with higher affinity to enter cells containing this surface receptor during infection [40]. Further, *Campylobacter jejuni* can mediate adherence to the intestinal epithelium through recognition of α−1,2-fucosylated carbohydrate moieties present on H blood group antigen, and human milk fucosyloligosaccharides can prevent or inhibit this binding [41]. As such, PL fimbriae represent a new mechanism for adherence of some *E. coli* to host tissues and cells, and recognition of specific α−1,2-fucosylated carbohydrates may provide an additional means for PL fimbriae-expressing *E. coli* to colonize extra-intestinal as well as intestinal mucosal surfaces in a variety of animal species. In this study, we also investigated binding specificity of the PlfG class I adhesin cloned from the UPEC strain UMEA3703–1, however, this adhesin did not recognize any specific sugars in our assays. Since adherence was maintained after treatment with metaperiodate, this variant of PL fimbriae may also bind to non-glycosylated receptors on cells or tissues. This demonstrates that PL fimbriae encoding either class I or class II adhesins exhibit distinct binding specificities.

The 3D structure of several fimbrial adhesins, including PlfG, were predicted using AlphaFold, with a focus on their lectin binding domains [42,43], and compared to crystallographic structures, such as FimH (type 1 fimbriae) and PapG (P fimbriae) [44–48]. We found that all fimbrial adhesins analyzed share a conserved β-sandwich fold within their lectin domains (Fig 8), suggesting an overall conserved mechanism of sugar recognition. Although the global folds are conserved, the glycan-binding pockets differ in size, shape, and charge, consistent with their distinct ligand specificities: FimH recognizes mannose, PapG binds Gal-α1–4-Gal, and PlfG targets fucose-α1–4-Gal. In PlfG adhesins, AlphaFold predictions indicate that the glycan-binding site contains hydrophilic and aromatic residues reminiscent of those found in the sugar-binding sites of both FimH and PapG [45,48]. However, despite the structural similarity of Class I and II PflG adhesins, it appears that their ligands are completely different nature, since the PlfG class I did not bind to any glycans present in the array and still mediated adherence to cells after treatment with sodium metaperiodate, which eliminates glycans from cell surfaces. Further understanding of how these adhesins bind different sugars or host receptors on cells and tissues may elucidate binding specificity that could lead to anti-adherence strategies to prevent or treat targeted infections. Future studies involving targeted mutagenesis and experimental structural characterization will be necessary to confirm the predicted binding sites and further dissect carbohydrate-binding specificity.

 

In conclusion, attachment of bacteria to host cells and tissues is often a prerequisite for colonization and subsequent infection. Herein, we have identified that PL fimbriae can contribute to virulence of avian pathogenic *E. coli* by promoting colonization of the lungs in turkey poults. Further, we identified that the PlfG adhesin mediates adherence to alpha1, 2-fucosylated carbohydrates. In a previous report, we determined these fimbriae were also present in some ExPEC strains associated with human infections, such as UTIs and bacteremia [19], and herein we also demonstrate that PL fimbriae also promote adherence to human uroepithelial cells. Taken together, these results support the potential role for PL fimbriae, in contributing to adherence and infection in both turkeys and humans. Considering the abundance of fucosylated glycans in the mucus of turkeys and the presence of Lewis B, Lewis Y, and H antigens on human O+ erythrocytes and epithelial cell surfaces, this supports a potential mechanistic explanation for PL fimbriae to contribute to cross-species potential of some *E. coli* strains associated with poultry products as a foodborne source with zoonotic risk for public health. PL fimbriae are encoded by a large transmissible virulence plasmid encoding several genes associated with ExPEC virulence, including *iroN*, *ompT*, *hlyF*, *iss*, *iutA*, *sitABCD*. As such, there is also a potential for the emergence of new *E. coli* pathotypes or variants through horizontal transfer when considering the important contribution of these plasmids to virulence and systemic infections caused by pathogenic *E. coli* affecting both poultry and mammals, including humans.

## Materials and methods

### Ethics statement

All animal procedures were conducted in accordance with institutional ethical guidelines and approved by the Comité Institutionnel de Protection des Animaux (CIPA) of the INRS–Armand-Frappier Santé Biotechnologie Research Centre. The protocol for air sac infection in turkeys was reviewed and approved under CIPA protocol number 2311–01. This study adhered to the ethical standards of the INRS campus and the National Experimental Biology Laboratory at INRS.

All blood procedures were conducted in accordance with institutional ethical guidelines and approved by the Institutional Committee on Research Ethics (CER) of the INRS–Armand-Frappier Santé Biotechnologie Research Centre. The protocol for blood collecting was reviewed and approved under CER protocol number 19–507. This study adhered to the ethical standards of the INRS campus and the National Experimental Biology Laboratory at INRS.

### Bacterial strains and growth conditions

Bacterial strains and plasmids are listed in Table 1. APEC strain QT598 and its derivative strains, clones of *E. coli* strain ORN172 expressing PL fimbriae, and UMEA-3703–1 (NCBI Bio sample no. SAMN01885978) isolated from the urine of a human with bacteremia, were grown at 37°C on solid or liquid Luria-Broth medium (Alpha Bioscience, Baltimore, MD, USA). When required antibiotic supplements were used at a final concentration of 100 µg/ml of ampicillin, 30 µg/ml of chloramphenicol, 100 µg/mL of carbenicillin, and 50 µg/ml of kanamycin. Brain heart infusion (BHI) broth (Alpha Bioscience, Baltimore, MD, USA) was used for the growth of QT598 and its derivatives prior to infection of turkey poults.

### Construction of the *Pplf-lux* reporter fusion

A transcriptional fusion between the *Plf* promoter and the *lux* operon genes was constructed by using Tn7 transposition [56]. The primers CMD2845 and CMD2846 were used for amplification of the promoter region by PCR and were ligated into pIJ514 [55] which had been digested with *XhoI* and *SpeI.* The ligation product was purified. Transformation into *E. coli* DH5α cells was followed by selection on LB plates containing chloramphenicol. The resulting plasmid pIJ610 was subsequently transformed into *E. coli* SM10 λ*pir*-derivative strain MGN-617. Strain MGN-617 (pGP-Tn7-*Pplf* promoter region-*lux*) was conjugated overnight with recipient strain QT598, containing plasmid pSTNSK, which encodes the Tn*7 tnsABCD* transposase genes, at 30°C on LB agar plates supplemented with DAP. After overnight conjugation with strain QT598, the bacteria from agar plates were suspended in 1 ml of phosphate-buffered saline (PBS), washed twice in PBS, serially

**Table 1. List of strains, and plasmids.**

| Strain, plasmid, or clone | Characteristic(s) | Reference |
|---|---|---|
| **Strains** | | |
| QT598 | APEC O1: K1 (Sequence type, ST1385) | [19] |
| *E. coli* MGN-617 | *thi thr leu tonA lacY glnV supE ΔasdA4 recA*:: RP42-Tc::Mu [pir]; Km r::RP4 | [49] |
| QT4420 | QT598 Δ*plf*::kan, Km r | [19] |
| QT2799 | *Serratia liquefaciens* | ATCC 27592 |
| MT78 | APEC O2:H+:K1, ST95 | [19] |
| UMEA-3703–1 | UPEC strain from the urine of a patient with bacteremia | [19] |
| QT6040 | MGN-617 (pIJ621); Gm r | This study |
| QT5953 | QT4420 (pSTNSK) Tp r | This study |
| QT6049 | QT5953:: *plf* Gm r | This study |
| QT5230 | CFT073 *fim* L-ON | [50] |
| QT5732 | ORN172/ pUCmT empty vector | [19] |
| ORN172 | *fim*-negative strain; *thr-1 leuB thi-1Δ(argF-lac)U169 xyl-7 ara-13 mtl-2 gal-6 rpsL tonA2supE44Δ(fimBEACDFGH)*::Km *pilG1* | [51] |
| QT6130 | QT598 containing GFP plasmid (pKEN), Carb r | This study |
| QT6131 | ORN172/pIJ497 (reference clone PL fimbriae with PlfGII adhesin), carries GFP plasmid (pKEN), Carb r | This study |
| QT6134 | QT598 Δ*plf*::kan, Km r, containing GFP plasmid (pKEN), Carb r | This study |
| QT5726 | ORN172/pIJ497 (reference clone PL fimbriae with PlfGII adhesin) | [19] |
| QT5814 | χ7213 + pGP-Tn7-Cm-P*plf* promoter region *luxCDABE*, Ap r, Cm r, Km r | This study |
| QT5828 | QT598::Tn7T-Cm::P*plf* promoter *luxCDABE*, Cm r | This study |
| QT4741 | ORN172/pIJ523 (reference clone expressing PL fimbriae with PlfGI adhesin) | [19] |
| **Plasmids** | | |
| pKD13 | Plasmid used for the amplification of the *kan* cassette | [52] |
| pKD4 | Plasmid used for the amplification of the *kan* cassette | [52] |
| pCP20 | FLP recombinase, Amp r | [52] |
| pSTNSK | Tn7 Transposase plasmid; pST6K*::tnsABCD*; ori*SC101*(Ts); Tp R | [53] |
| pIJ621 | pGP-Tn7-Gm::*plf* Gm r, Amp r | This study |
| pIJ497 | pUCmT::*plf*QT598 | This study |
| pGp-Tn7-Gm | Cloning vector for *att* Tn7 complementation | [53] |
| pUCM-T | Cloning-vector; Amp R | Bio Basic Inc., Markham, ON., Canada |
| pKEN | pKEN GFP mut2 | [54] |
| pIJ523 | pBC-SK+ contains P-like fimbriae of UMEA | [19] |
| pIJ514 | pGP-Tn7-Cm::*luxCDABE*; *rbs* Ap r, Cm r | [55] |
| pIJ610 | pGP-Tn7-Cm::P*plf* promoter region *luxCDABE*, Ap r, Cm r | This study |

diluted, and cultured on LB agar supplemented with Chloramphenicol, and incubated at 37°C. The Chloramphenicol-resistant colonies that grew were then tested for sensitivity to kanamycin and ampicillin, indicating the likelihood of integration at *att*Tn7 and loss of the transposase-encoding plasmid *pSTNSK*. Insertion of Tn*7* into the *att*Tn7 site was verified by PCR with primers CMD26 and CMD1416, which are listed in Table 2.

## Deletion of *plf* genes from QT598 and complementation of the Δ*plf* mutant

The *plf*-negative mutant strain QT4420 was constructed by the lambda red recombinase method by deleting the gene cluster (*plfBAHCDJKEFG*) [19]. The *plf* gene cluster (*plfBAHCDJKEFG*) was amplified by PCR using primers (Table 2) and Phusion Flash high-fidelity PCR master mix (Thermo Fisher Scientific, Waltham, MA, USA) from plasmid DNA of pIJ497 for the complementation. The insert segment amplified by PCR contained Gibson homologies to the *XhoI* and *StuI* sites within the multi-cloning site (MCS) of plasmid pGP_miniTn7_Gm for *plf*. Plasmid pGP_miniTn7_Gm was linearized by restriction digestion using enzymes *StuI* and *XhoI* (New England BioLabs, Ipswich, MA, USA). Plasmid pIJ621 was generated by cloning the DNA fragment containing the *plf* genes into linearized pGP_miniTn7_Gm, using the pEASY-UNI assembly kit (TransGen Biotech, Beijing, China). The construct was then transformed into *E. coli* strain MGN-617 as a donor strain for conjugation into the Δ*plf* mutant containing the Tn7 transposase-encoding plasmid pIJ360, as a recipient strain, to construct QT598Δ*plf* strain (QT4420). Conjugation was performed using the method described by Crépin *et al.* 2012 [53].

## Protein preparation and western blot analysis

Fimbriae and other surface proteins were extracted from the desired strains: QT598, QT598 Δ*plf* (QT4420)*,* and the *plf* complemented strain (QT6049) based on the heat extraction method as described previously, with some modifications [19]. The overnight cultures were incubated at 55 °C for 1h, and supernatants were harvested by centrifugation at 6000 x *g* for 15 minutes. For precipitation of the Plf proteins, 15% trichloroacetic acid (TCA) (v/v) was used, and then the samples were incubated at 4 °C for 1 h with shaking. The precipitated proteins were then concentrated by centrifugation at 14000 x *g* for 20 min at 4 °C. Samples were then washed twice with Tris-EDTA (0.05 M) pH 12, then once with Tris-EDTA (0.05 M) pH 8.5; and resuspended in 0.1 ml of Tris-EDTA (0.05 M) pH 8.5 [19,49,57]. The loading samples were prepared using 4x Laemmli sample buffer (Tris-HCL 200 mM, 8% SDS (v/v), 40% glycerol (v/v), 4% β-mercaptoethanol (v/v), 50 mM EDTA,

**Table 2. Lists of primers used in this study.**

| Primer name | Direction | Characteristic(s) | Sequence 5'-3' |
|---|---|---|---|
| CMD2852 | Forward | Gibson pairs to amplify the *plf* gene with the promoter from pIJ497 to clone it into the *stuI* site of pGP-Tn7-gm. (Used with CMD2853) | TTGGGCCCGGTACCTCGCGAAG-GCTTCGGAGATCAAGACACTG |
| CMD2859 | Reverse | Gibson pairs to amplify *plf* gene with the promoter from pIJ497 to clone it into the *XhoI* site of pGP-Tn7-gm. (Used with CMD2858) | TACCGGGCCCAAGCTTCTC-GACTCCAGTCTTATGAACGGGC |
| CMD2845 | Reverse | Gibson pairs to amplify *plf* region with promoter from QT598(used to clone *plf* into *XhoI* site of pGP-Tn7-vector) | ATGGGGGCCCACCTCCTCGAGC-TACGCCAAGAACCGATTTC |
| CMD2846 | Forward | Gibson pairs to amplify *plf* region with promoter from QT598(used to clone *plf* into *SpeI* site of pGP-Tn7-vector) | ATCATGCATGAGCTCACTAGTAG-CACACGCACTGTTTATGG |
| CMD26 | Forward | In *glmS* for screening integration in *att*Tn7 site (used with CMD1416) | GATCTTCTACACCGTTCCGC |
| CMD1416 | Reverse | In *glmS* for screening integration in *att*Tn7 site | GCTTTTTCACAGCATAACTGGA |
| CMD2186 | Forward | qRT-PCR *plfA* | CGGATCAGGGACAAGGTAAAG |
| CMD2187 | Reverse | qRT-PCR *plfA* | CAGCCAGATGAGCTTTGG |
| CMD2904 | Forward | Screen primer for *plf* in pGP-Tn7-gm | AAGGGAATCAGGGGATCT TG |
| CMD1421 | Reverse | Screen primer for *plf* in pGP-Tn7-gm | AACCGTATTACCGCCTTTGA |

and 0.4% bromophenol blue (v/v) and then 10 µg of proteins were migrated and resolved on 15% SDS-PAGE gels at 200 V for 40 min. The proteins were transferred to a nitrocellulose membrane (Pall Corporation, Port Washington, USA). Membranes were then placed in tris-buffered saline (TBS) with 0.1% Tween-20 (TBST), and 5% (w/v) skim milk was used to block the membrane for 1h at room temperature with shaking [19,49]. The membrane was incubated overnight at 4 °C, in TBST containing 5% skim-milk, with rabbit polyclonal antibodies provided by New England Peptide (1:1,000) against a peptide corresponding to part of the PlfA major fimbrial subunit (Ac-CAHLAADGISVKKD-amide). The membrane was then washed three times using TBST and incubated with secondary antibody anti-Rabbit HRP diluted (1:10000) (Novus), in TBST for 1 h at room temperature, and washed three times using TBST [19]. Signals were then revealed using SuperSignal West Pico PLUS Chemiluminescent Substrate (Thermo Fisher Scientific, Waltham, MA, USA) and visualized with the ChemiGenius 2 documentation system according to the manufacturer's instructions.

## Turkey infection model and sample collection

All animal procedures followed institutional guidelines and were approved under protocol CIPA No. 2311–01 by the "Comité Institutionnel de Protection des Animaux". One-day-old turkey poults, with equal numbers of males and females, were obtained from a local commercial hatchery in Quebec and were transported to the National Experimental Biology Laboratory at the INRS campus in Laval, Quebec. Birds were housed in isolators with no more than 10 poults per group. On day 6 post-hatch, poults were challenged by direct injection into the left air sac with either the wild-type *E. coli* strain QT598, the QT598 Δ*plf* mutant, or the complemented *plf* strain. Each turkey received $3.5 \times 10^9$ CFU in 100 µL of inoculum. Following infection, poults were monitored daily for 48 hours for clinical signs such as lethargy, changes in posture, eye condition, and respiratory distress. At 48 hours post-challenge, poults were euthanized and necropsied to evaluate macroscopic lesions in the air sacs, heart, and liver. Fibrinous lesions were scored, and tissue samples including the right air sac, right lung, spleen, and liver were collected aseptically, weighed, buffered saline gelatin (BSG: 8.5g/L NaCl, 0.3g/L KH$_2$PO$_4$, 0.6g/L Na$_2$HPO$_4$, 0.1g/L gelatin) was added in a amount of twice the tissue weight, and then homogenized. Tissue homogenates were serially diluted in sterile BSG and plated on MacConkey-lactose agar. Plates were incubated at 37 °C for 16–18 hours to quantify bacterial colonization. Heart blood was also collected at 48 hours post-infection and plated for quantification of bacteria. For all CFU counts and weight change data, the distribution of values was assessed using the Shapiro-Wilk test. For datasets that met normality assumptions, one-way ANOVA followed by Tukey's post-hoc test was applied. The number of poults per group was n = 10, and exact P values for each comparison are provided in the figure legends.

## qRT-PCR analysis of *plf* gene expression *in vivo* and *in vitro*

For *in vitro* analysis, *plf* expression was assessed after growth of the QT598 strain after 3 passages on LB agar (LBA) at 37 °C by analysing the RNA transcript level of *plfA* encoding the major subunit of PL fimbriae. For *in vivo* analysis, total RNA was extracted from the infected turkey lungs 48 h post-infection. RNA was extracted using a EZ-10 Spin Column Total RNA Miniprep Kit (BioBasic) according to the manufacturer's protocol after adding RNAprotect (Qiagen) to bacterial cultures. Then, Ambion Turbo DNase (Thermo Fisher Scientific) was used to eliminate contaminating DNA from the samples, and the eluted RNA samples were confirmed by the detection of the *rpoD* gene by PCR (40 cycles). Turkey lung samples were collected from 5 poults infected with QT598, and tissues were then homogenized with TRIzol reagent (Thermo Fisher Scientific), centrifuged for 30 sec at 12,000 × g, and the supernatant was incubated with ethanol (95–100%) and transferred into Zymo-Spin IICR Columns (Zymo Research). A260/A280 Nanodrop readings and agarose gel electrophoresis were used to evaluate the RNA concentrations and integrity. Total RNAs were then reverse transcribed to cDNAs using *TransScript* all-in-one first-strand cDNA synthesis supermix kit (TransGen Biotech Co., Ltd, Beijing, China) according to the manufacturer. Specific primers (Table 2) against *plfA* and *rpoD* as a housekeeping control were used. qRT-PCR was performed in the Corbett instrument with reaction mixtures containing 50 ng of cDNA, 100 nM of each primer, and 10µl of *TransStart* tip

green qPCR supermix (TransGen Biotech Co., Ltd, Beijing, China). Data were analyzed using the 2−ΔΔCT method [58]. Genes with a fold-change above a threshold of 2 were considered differentially expressed.

## Bacterial adherence to turkey and chicken lung frozen sections

Six-day-old turkey poults and six-day-old chicks were obtained from a commercial hatchery and transported to the National Experimental Biology Laboratory at the INRS campus in Laval, Quebec. Whole lungs were immersed in Optimal Cutting Temperature compound (OCT) embedding matrix (VWR, USA) to slowly freeze. Then, they were kept at -80 °C prior to slicing with a microtome. Tissue blocks were placed in the cryo-microtome chamber and allowed to reach -20 °C, then sections with a thickness of 5 µm were prepared using a cryostat (Leica CM1850 Cryostat, Leica Biosystems, Germany). The tissues were placed on a positively charged glass slide on top of the section. The tissue was adhered to the slide and air-dried for 30–60 min [57,59].

Bacterial strains containing a GFP-expressing plasmid, pKEN, were grown in LB overnight and on LB agar for 3 passages at 37°C. For each of the samples, bacterial pellets were obtained by centrifugation at 4000x $g$ for 6 min, followed by washing 2 times with PBS and diluted for a final bacterial cell count of ~$1 \times 10^{8}$ CFU/mL. Bacteria with GFP were visualized by confocal microscopy (Thermofisher, USA, excitation 488 nm, emission 509 nm). For adherence inhibition assays, the GFP-labeled strains were pre-incubated with sugars for 30 min at 4 °C prior to interaction with lung tissue sections. All the pictures were magnified 120x and the experiments were repeated 3 times. The nuclei of lung sections were then stained with 4′,6-diamidino-2-phenylindole (DAPI) 1µg/ml (Thermofisher, USA, excitation 358 nm, emission 461 nm) for 5 min at room temperature in the dark, followed by 3 washing steps with PBS. After that, to stain the membrane of the lung sections, WGA-Alexa Fluor 647 (Thermofisher, USA, 5 µg/ml in PBS, excitation 650, emission 665 nm) for 5 min at room temperature in the dark, followed by 3 washing steps with PBS. Finally, 10 µL of each stained strain was added to the slides, separately, and incubated on ice in the dark for 30 min. The slides were washed with PBS 3 times to wash away unattached bacteria. Slides were fixed with 2% paraformaldehyde (PFA, E. Merck AG, Darmstadt, Federal Republic of Germany) for 10 min on ice, followed by 3 washing steps with PBS, and a coverslip was placed on them for visiualization by confocal microscopy. Confocal images were acquired using a Zeiss LSM 780 (Carl Zeiss, Germany) with sequential laser excitation for DAPI, Alexa Fluor 555, and GFP. Z-stack images were collected through the epithelial surface of OCT-embedded lung tissue. ZEN Black software (version 2012 Sp4; Zeiss) was used for image acquisition, channel separation, and intensity calibration [60,61].

## Hemagglutination assays

Hemagglutination tests were performed in the presence of 2.5% D-mannose, to eliminate agglutination due to type 1 fimbriae. Turkey and human O+ erythrocytes were tested on depression glass slides. The centrifugation and washing of the erythrocytes were repeated twice. The 3% final concentration of erythrocytes was prepared by suspension in phosphate-buffered saline (PBS). Strains such as QT5726 (clone producing PL fimbriae with PlfG class II adhesin from QT598), QT4741 (clone producing PL fimbriae with PlfG class I adhesin from UMEA-3703–1), QT598, QT4420 (QT598Δ*plf*), and QT6049 (complemented *plf*) were grown on LB agar until the 3rd passage at 37 °C. The strains were then centrifuged at 3,000× $g$ for 15 min. Then, pellets were suspended in PBS (pH 7.4) and adjusted to an optical density at 600 nm ($OD_{600}$) of 0.6, centrifuged, and then concentrated 100-fold in PBS (pH 7.4). The agglutination was determined after 30 min of incubation on ice, visually and graded as -, +, ++, or +++ [15,62]. Sodium metaperiodate (Thermofisher, USA) was used to determine whether the binding of strains required sugar-specific receptors. The RBCs were treated with various concentrations of sodium metaperiodate on ice, and then the strains were added to visualize any hemagglutination [63].

## Hemagglutination inhibition tests

Hemagglutination inhibition (HAI) tests were performed with turkey or human O+ erythrocytes with various concentrations of sugars (monosaccharides, disaccharides, and complex sugars). Sugars were dissolved in PBS from 2M to 0.5mM final

concentrations. The centrifugation and washing of the erythrocytes were repeated twice. First, in V-shaped 96-well microtiter plates, 30 µL serially diluted sugar solutions were mixed with 30 µL of the bacterial suspension for 15 min; afterwards, a 3% suspension of each erythrocyte (30 µL) was added. The plates were incubated on ice for 30 min, and the assay was visually inspected. The lowest sugar concentration that inhibited hemagglutination was determined as the inhibition titer (IT) [15,16,64,65]. Hemagglutination inhibition (HAI) was also performed using Lewis B and H antigen sugars (Biosynth, the United Kingdom) with washed human O+ erythrocytes and QT5726 (ORN172 expressing PlfG class II cloned from strain QT598), separately. The Lewis antigen was dissolved in 50 µL PBS to yield a stock concentration of ~37.8 mM. Then the erythrocytes, bacteria, and sugars were combined on a glass slide on ice for 30 min. The agglutination or inhibition was visualized by the light microscope.

## Adherence assay with and without the addition of inhibitor sugars

The kidney HEK-293 (ATCC CRL-1573) epithelial cell line was grown to confluence in 24-well plates in Eagle's minimal essential medium (EMEM) (Wisent Bio Products, St-Bruno, Canada) supplemented with 10% fetal bovine serum (FBS) at 37°C in 5% $CO_2$ to confluency, and then $2 \times 10^5$ cells/well were distributed in 24-well plates. Wild-type strain QT598 and its derivatives were grown on LB agar until the 3rd passage at 37 °C and other strains were grown in LB broth overnight. For all the strains mannose was added to inhibit type 1, except the positive control strain MT78. Strains were then centrifuged at $3,000 \times g$ for 15 min. Pellets were then suspended in PBS (pH 7.4), adjusted to $10^6$ CFU ml$^{-1}$, and incubated with and without inhibitor sugars at a concentration of 50 mM for 30 minutes on ice. Immediately before interaction with human cells, cultures were washed once with PBS and seeded on host cells at an estimated MOI of 10 CFU per cell. Bacteria-host cell contact was enhanced by a 5-min centrifugation at $600 \times g$. After 2 h, cells were washed three times and lysed with PBS–0.1% sodium deoxycholate (DOC), serially diluted, and plated on LB agar plates. Quantification of cell-associated bacteria was performed. To inhibit adherence mediated by type 1 fimbriae, 2.5% α-d-mannopyranose was added to the culture medium.

## Glycan array tests

Glycan microarray (Glycan Array 100) slides were purchased from Ray Biotech (Peachtree Corners, GA, USA) [33–35,66]. Bacterial strains such as QT5726 (ORN172 PlfG class II), QT4741 (PlfG class I adhesin cloned from strain UMEA-3703–1), QT5230 (CFT073 *fim* L-ON), and QT5732 (ORN172) were grown on LB at 37 °C. Fresh cultures of each strain were prepared using LB at 37 °C with shaking at 250 rpm (OD600 = 0.6). For each sample, supernatants were obtained by centrifugation at $4000 \times g$ for 6 minutes, followed by three washes with PBS. Next, 20 µL of Dimethyl Sulfoxide (DMSO) was added to a vial of CellTrace Far Red staining solution (Thermofisher, USA, excitation 640 nm, emission 660 nm), and then it was diluted in 20 mL PBS to make a 1 µM staining solution. Cells were resuspended in 10 mL Cell-Trace Far Red staining solution, incubated at 37 °C with shaking at 250 rpm for 30 min. Strains were diluted 1:1000 in PBS buffer for a final bacterial cell count of ~$1 \times 10^8$ CFU/mL [33]. The glycan slide was equilibrated to room temperature inside a sealed plastic bag for 20–30 minutes, then the bag was removed, and slides air-dried at room temperature for another 1–2 h. Each well on the glass slide was blocked by adding 400 µL of blocking buffer and incubating at room temperature for 30 minutes. The buffer was decanted from each well, and 400 µL of each strain was added to the corresponding well. The array was incubated with shaking at 4 °C for 2 hours. Each well was washed three times with PBS with shaking, followed by washing the glass slide with distilled water for 5 minutes. Finally, droplets were removed by suction with a pipette. Stained bacteria were detected using a confocal microscope.

## Structural modelling

Structural modelling and comparative analysis of fimbrial adhesins were done using the AlphaFold multimer server to predict the monomeric arrangement of the PlfG class I and II adhesin (full-length domains). For each prediction, all five generated models converged on the same structural fold, yielding pTM scores of 0.67 and 0.69 for PlfG class I and II,

respectively [36,42,43]. The corresponding predicted local distance difference test (pLDDT) scores and predicted aligned error (PAE) plots are provided in the supplementary information as S10 Fig. To assess structural conservation and functional divergence, the PlfG class II adhesin was compared to crystallographically resolved fimbrial adhesins with known sugar-binding specificities: PapG-II (P fimbriae; binds tetrasaccharides, PDB: 4Z3E), and FimH (Type 1 fimbriae; binds mannose, PDB: 1KLF) [20,21,44–47,67]. Bound ligands present in the crystal structures were retained to analyze conserved interaction networks. Residues implicated in glycan recognition were compared across adhesins, and potential binding residues in PlfG were annotated based on structural homology and AlphaFold confidence metrics. PyMOL was used to generate figures, with carbohydrate-binding pockets and key residues highlighted.

## Binding-site comparison

Carbohydrate-binding pockets were identified based on previously published crystal structures and AlphaFold-predicted models [68]. Key residues involved in ligand recognition (e.g., Y48, N135 in FimH and PapG) were compared across adhesins using distance measurements and hydrogen bond analysis in PyMOL. Comparative binding groove features were also assessed qualitatively.

## Statistical analyses

All experimental data are presented as mean ± standard error of the mean (SEM). For comparisons between two groups, a two-tailed Student's t-test was applied. When comparing more than two groups, one-way analysis of variance (ANOVA) was employed. In the case of turkey infection experiments, pairwise comparisons were analyzed using the Mann–Whitney test, while group comparisons were evaluated with the Kruskal–Wallis test. A P-value less than 0.05 was considered indicative of statistical significance. All analyses were conducted using GraphPad Prism version 7 (GraphPad Software, San Diego, CA, USA).

## Supporting information

**S1 Fig. Stability of *plf* promoter activity across passages under different growth conditions.** Luminescence from strain QT598 carrying the *pPlf-lux* reporter was measured over three consecutive passages in LB broth cultured without agitation (left) and with agitation (right) at 37°C. Under static conditions, luminescence remained stable across all passages with no significant difference (ns). Data represent mean ± SEM of biological replicates.
(PDF)

**S2 Fig. Increased Activity of *plf* promoter with consecutive passages on LB agar.** Luminescence from QT598 *pPlf-lux* was measured over three consecutive passages of growth on LB agar. The luminescence levels showed significant increases between passages. Data are the means from three independent experiments, and error bars represent standard errors of the means. *, $P < 0.05$; **, $P < 0.01$; and ***, $P < 0.001$, ****, $P < 0.0001$ using one-way ANOVA.
(PDF)

**S3 Fig. Macro-hemagglutination (HA) inhibition test of human O⁺ erythrocytes by PL fimbriae-producing strains in the presence of sodium metaperiodate.** Positive control is HA with strain QT4741 (PlfG class I adhesin cloned from strain UMEA-3703–1). Hemagglutination was not inhibited at concentrations ranging from 0.03 M to 0.1 M sodium metaperiodate, whereas blood was lysed at higher concentrations. Agglutination inhibition was visiualized after 30 min of incubation on ice.
(PDF)

**S4 Fig. Macro-hemagglutination (HA) inhibition test of human O⁺ erythrocytes by PL fimbriae-producing strains in the presence of sodium metaperiodate.** Positive control is HA with strain QT5726 (Clone PlfG class II from strain

QT598), and negative control is QRN172. Hemagglutination was inhibited at concentrations ranging from 0.03 M to 0.1 M sodium metaperiodate, whereas blood was lysed at higher concentrations. Agglutination inhibition was visiualized after 30 min of incubation on ice.
(PDF)

**S5 Fig. Macro-hemagglutination (HA) inhibition and micro-hemagglutination inhibition tests of turkey and human erythrocytes by PL fimbriae-producing strains in the presence of various sugars.** A) Positive control is HA with strain QT5726 (Clone PlfG class II from strain QT598), L-fucose and D-galactose were shown to inhibit HA by strain QT598, the negative control is QT598Δ*plf*, and the other sugars can not inhibit the hemagglutination. B) Micro-hemagglutination inhibition test of turkey and human red blood cells. Positive control is HA with strain QT5726 (Clone PlfG class II from strain QT598), L-fucose and D-galactose were shown to inhibit HA by strain QT598, the negative control is QT598Δ*plf.* Agglutination inhibition was visiualized after 30 min of incubation on ice.
(PDF)

**S6 Fig. Adherence inhibition of Plf-expressing strains to HEK 293 kidney cells in the presence of inhibitor sugars.** Monolayers were infected for 2 h, and adherent bacteria were quantified by plating on LB Agar. Data are expressed as CFU/mL; bars represent means ± SEM from three independent experiments. QT5726 (ORN172 expressing PL fimbriae with PlfGII adhesin) ****$P < 0.0001$; ***$P < 0.001$; ns, not significant.
(PDF)

**S7 Fig. Arrangement of glycans in each subarray of the glycan array slide.** (A) Each slide is divided into four subarrays, with 100 distinct glycans immobilized on each subarray. (B) The table presents the spatial arrangement and numbering of glycans. For example, position 3 corresponds to mannose, position 61 to the H antigen, position 63 to the Lewis Y, and position 64 to the Lewis B. A complete list of glycans and their corresponding positions can be found on the RayBiotech Glycan Array 100 webpage.
(PDF)

**S8 Fig. Hemagglutination inhibition (HAI) test of QT5726 with Lewis B antigen and H antigen with O + RBCs.** Positive control is strain QT5726 (ORN172 expressing PlfG class II cloned from strain QT598), negative control is ORN172. Addition of Lewis B or H antigen resulted in inhibiton of HA. Hemagglutination inhibition was scored visually after 30 min of incubation on ice.
(PDF)

**S9 Fig. The amino acid alignment between PlfG class II and PlfG class I.** The common amino acids are indicated in red between PlfG class II from QT598 and PlfG class I from UMEA 3703–1.
(PDF)

**S10 Fig. AlphaFold model generation.** AlphaFold confidence metrics, including predicted aligned error (PAE), predicted local distance difference test (pLDDT), as well as and predicted TM-score (pTM) are shown for A) the PlfG QT598 and B) PlfG UMEA3703–1 models.
(PDF)

## Acknowledgments

We thank Jessy Tremblay for assistance with confocal microscopy and Prof. Isabelle Plante for assistance with the cryo-microtome.

## Author contributions

**Conceptualization:** Fariba Akrami, Paula Armoa Ortiz, Charles Calmettes, Charles M. Dozois.

**Data curation:** Fariba Akrami, Paula Armoa Ortiz, Charles Calmettes.

**Formal analysis:** Fariba Akrami, Sebastien Houle, Paula Armoa Ortiz, Charles Calmettes.

**Funding acquisition:** Charles Calmettes, Charles M. Dozois.

**Investigation:** Fariba Akrami, Hossein Jamali, Sebastien Houle, Charles M. Dozois.

**Methodology:** Fariba Akrami, Hossein Jamali, Sebastien Houle, Paula Armoa Ortiz, Charles M. Dozois.

**Project administration:** Charles M. Dozois.

**Resources:** Charles Calmettes, Charles M. Dozois.

**Software:** Fariba Akrami.

**Supervision:** Sebastien Houle, Charles Calmettes, Charles M. Dozois.

**Validation:** Fariba Akrami, Paula Armoa Ortiz, Charles M. Dozois.

**Visualization:** Fariba Akrami.

**Writing – original draft:** Fariba Akrami, Charles M. Dozois.

**Writing – review & editing:** Fariba Akrami, Hossein Jamali, Sebastien Houle, Paula Armoa Ortiz, Charles Calmettes, Charles M. Dozois.

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
