## [Decision Letter · Decision Letter 0]

23 Dec 2025

PPATHOGENS-D-25-02746

Host range and zoonotic potential linked to P-like fimbrial adhesin specificity in avian pathogenic Escherichia coli

PLOS Pathogens

Dear Dr. Dozois,

Thank you for submitting your manuscript to PLOS Pathogens. After careful consideration, we feel that it has merit but does not fully meet PLOS Pathogens's publication criteria as it currently stands. Therefore, we invite you to submit a revised version of the manuscript that addresses the points raised during the review process.

We look forward to receiving your revised manuscript.

Kind regards,

Eric Oswald, Ph.D., D.V.M.

Academic Editor

PLOS Pathogens

Thomas Guillard

Section Editor

PLOS Pathogens

Sumita Bhaduri-McIntosh

Editor-in-Chief

PLOS Pathogens

orcid.org/0000-0003-2946-9497

Michael Malim

Editor-in-Chief

PLOS Pathogens

orcid.org/0000-0002-7699-2064

**Journal Requirements:**

At this stage, the following Authors/Authors require contributions: Hossein Jamali, Sebastien Houle, Paula Armoa Ortiz, Charles Calmettes, and Charles M. Dozois. Please ensure that the full contributions of each author are acknowledged in the "Add/Edit/Remove Authors" section of our submission form.

- ® on pages: 34, and 35

- TM on pages: 34, and 36.

Potential Copyright Issues:

i) Please confirm (a) that you are the photographer of 1, 5, S3, S4, S5, and S8, or (b) provide written permission from the photographer to publish the photo(s) under our CC BY 4.0 license.

**Reviewers' Comments:**

Reviewer's Responses to Questions

**Part I - Summary**

Reviewer #1: This study attempts to determine the role of P-like (PL) fimbriae of APEC strain QT598 in a turkey respiratory infection model. The manuscript is interesting because it shows that PL fimbriae binds to alpha1,2 fucosylated carbohydrates, representing a new mechanism promoting adherence of E. coli to host tissues and cells. The experiments conducted and the conclusions drawn are convincing.

Reviewer #2: This manuscript presents a solid and compelling study that identifies PL fimbriae encoded on a ColV plasmid in the APEC strain QT598, demonstrates their contribution to turkey lung colonization and tissue tropism, and defines the glycan-binding specificity of the PlfG class II adhesin. The authors show that PL fimbriae preferentially recognize Fuc-α1,2-Gal–containing Lewis antigens—specifically Lewis b, Lewis y, and H structures—offering mechanistic insight into receptor usage on avian respiratory tissues. The finding that these same glycans are also present on human epithelial cells raises important questions about potential cross-species interactions and zoonotic risk, providing a conceptual link between APEC virulence factors and host range expansion. Overall, the work is well conceived and fits within the scope of PLOS Pathogens, as it connects a defined virulence determinant to mechanisms of colonization, host specificity, and possible zoonotic potential. The study advances our understanding of PL fimbriae biology and offers a framework for exploring receptor-driven host tropism in ExPEC strains.

Strength of evidence for virulence in the turkey model

1. The turkey air sac model is a strong point and biologically relevant for this APEC strain (QT598 was originally isolated from a turkey).

2. Deletion of plf leads to a 2–3 log₁₀ reduction in lung CFU, which is nicely complemented by chromosomal reintroduction of plf (Fig 2A, pp. 9–10). However: a) colonization differences in liver and spleen are modest and not statistically significant (Figs 2B, 2C). The text should avoid over-interpreting systemic spread and keep the claims focused on lung colonization; b) there are no clinical signs of colibacillosis, and the main host effect is altered weight gain (Fig 2D). The authors should better frame this as subclinical impact on host fitness, not as severe systemic disease.

The manuscript links glycan specificity (Lewis b, Lewis y, H) to a possible zoonotic/foodborne risk from poultry to humans (Abstract and Discussion, pp. 2, 25–27). Evidence for human relevance in this paper: a) PL fimbriae promote adherence to HEK293 human kidney cells (Fig 6, p. 18–19); b) Glycan arrays show binding to Fuc-α1,2-Gal–containing structures present on human O⁺ erythrocytes and some mammalian epithelia; c) Prior work (cited) shows PL fimbriae in some human ExPEC. However, direct zoonotic transmission is not demonstrated here. No in vivo mammalian model or epidemiological link is included in this manuscript. Therefore I recommend that the authors soften the language about “zoonotic and foodborne risk” and frame it as “potential for cross-species colonization based on shared glycan receptors” rather than a demonstrated zoonotic pathway.

The introduction requires revision, as it states the incorrect concept that ExPEC belongs to the human intestinal commensal microbiota; in fact, ExPEC strains are pathobionts rather than true commensals (line 88).

Reviewer #3: The study entitled “Host range and zoonotic potential linked to P-like fimbrial adhesin specificity in avian pathogenic Escherichia coli” by Akrami and co-authors investigate contribution of plasmid-encoded P-like fimbriae in colonisation of turkey lungs and glycan moieties that mediate binding of Plf to host cells. While the work addresses important questions regarding host range and zoonotic potential, the current organization of the manuscript makes the central message difficult to discern. To maximize the impact of the findings, I strongly recommend streamlining the narrative. The flow of the results is currently interrupted by sporadic comparisons with UPEC strains, description of methodology and changes in strains used during the manuscript. Changing the manuscript to first focus specifically on the turkey-associated Plf variant (Class II) would create a much more cohesive and compelling story. Additionally, the figures require optimization to improve readability and resolution, which is essential for supporting the text.

Reviewer #4: This is a highly interesting manuscript that describes the characterisation of the ColV-plasmid encoded P-like (PL) fimbriae from avian pathogenic E. coli (APEC). The authors show PL fimbriae contribute to colonisation of the lungs in a turkey infection model, and provide additional data to demonstrate species- and tissue-specific tropism. The expression of plf genes was found to be upregulated in the lungs of infected turkeys using qRT-PCR. Glycan array analysis revealed the receptor target profile of the PlfG adhesin, and these findings were supported by structural modelling.

I thoroughly enjoyed reading this manuscript and could not fault the work. The data presented are sound, the manuscript is very well written, and the work is of significant interest to the global one health problem of bacterial pathogenesis and antibiotic resistance.

**Part II – Major Issues: Key Experiments Required for Acceptance**

Please use this section to detail the key new experiments or modifications of existing experiments that should be absolutely required to validate study conclusions.required to validate study conclusions.

Reviewer #1: No major issue.

Reviewer #2: Concerning the results and discussion:

1. The confocal images comparing adherence to turkey vs chicken lung sections (Fig 4, pp. 12–13) are compelling visually, but: a) the conclusions rely heavily on representative images; b) quantitative analysis (e.g., number of GFP⁺ bacteria per unit area, fluorescence intensity per field) would make this much stronger. Therefore I suggest some quantification of adherence on tissue sections (even semi-quantitative, blinded counts in multiple fields) to support the claim of species-specific tropism.

2. The glycan array data (Fig 7, pp. 19–20) are a major strength and a key novelty. Thus, some points to clarify / strengthen:

a) Specificity for Lewis antigens and H antigen; b) They nicely show binding to Lewis b, Lewis y, and H, and inhibition by L-fucose and D-galactose in HA and tissue assays (Fig 5); c) It would be good to clearly state in the Results and Discussion whether any other glycans showed meaningful binding above background, or if binding was truly restricted to those Fuc-α1,2-Gal structures.

3. Relevance to turkey lung

a) In the discussion the authors cites literature indicating fucosylated glycans in avian respiratory mucins, and reduced Lewis antigens in chicken lungs, but, there is no direct demonstration in this manuscript that Lewis b/y or H antigens are present on turkey lung sections (e.g., lectin or antibody staining). Therefore. I suugest to the authors to discuss this limitation explicitly and avoid over-stating that the identified glycans are definitively present on turkey lungs; instead phrase as “consistent with reported fucosylated structures in avian respiratory mucosa”; but, iof the data already exist (e.g. lectin histochemistry), they should be added or referenced clearly.

4. In HEK293 adherence assays, addition of L-fucose or D-galactose reduces binding of plf⁺ strains (Fig 6 and Fig S6, p. 18–19).

Points you might raise: a) clarify which blood group / glycan profile HEK293 cells have, or at least discuss this as a caveat; b) ensure that sugar concentrations and controls (osmotic effects, toxicity) are clearly described in Methods.

5. The AlphaFold-based structural modeling (Fig 8, pp. 21–22) is useful but somewhat descriptive: a) it supports a conserved β-sandwich fold and shows plausible residues for fucose/galactose binding; b) it remains predictive and not experimentally validated. Thus, I suggest to the authors to explicitly state that structural insights are predictions and to avoid language that implies experimentally proven binding residues. A short paragraph proposing future mutagenesis/structural studies would strengthen this section.

6. Statistics and sample size

a) For animal experiments (colonization, weight gain): make sure the number of poults per group (n) is clearly given in figure legends (Fig 2) and that the statistical tests (ANOVA, post-hoc) are appropriate.

b) Clarify whether data distribution was checked and whether non-parametric tests might be needed for CFU counts (often not normally distributed). Please provide exact n, P values, and statistical tests for each comparison in the figure legends and Methods.

Reviewer #3: 1. Lack of assays with tissue section from different organs.

2. Lack of qPCR results with expression data for Pfl adhesin in different tissues/gut.

3. Lack of quantitative data for haemagglutination and tissues section results.

Reviewer #4: I have no substantive concerns regarding the data or major conclusions.

**Part III – Minor Issues: Editorial and Data Presentation Modifications**

Reviewer #1: General comments :

The names of the strains are different from one section and/or figure to another. Please harmonize for clarity.

Example no 1: QT5726 (Fig S6) versus ORN172 PlfGII (Fig S5)

Example no 2: QT5741(text L292) versus ORN172 PlfGI (Fig S3)

L160. Results for turkey erythrocytes could not be reviewed as they are not provided.

L303-308. Please show the results of experiments with different concentrations of fucose and galactose.

L530-533. It was not demonstrated that receptors of Plf were low or absent on chicken tissues or cells. Only reduced binding was shown. Please re-write.

Specific minor points :

L71. Delete “also”

L77-79. This sentence is identical to that found in L48-50.

L113. Please add the year of isolation, disease caused by QT598, phylogenetic group and Sequence Type. Provide also a reference.

L116. Could you please provide some details about the prevalence of plf genes in UPEC and APEC ? Please provide also a reference.

L120. Fig S9 could be mentioned here.

L131. Please add “respectively”

L153. Please delete ‘with primers”

L181. CFU/ml of tissue.

L226 (and elsewhere). “in vitro” and “in vivo” should be italicized.

L264. In A and B, please add “with turkey lung” (as in E and F)

L270. It would have been relevant to complement QT598 delta plf to confirm that binding to lung tissue was restored.

L276-277. Please indicate Class II for Plf in QT6131.

L289. Please indicate the role of sodium metaperiodate.

L291. Please delete “Fig S3” as this figure does not contain results from QT598.

L294. Please add “Fig S3”.

L297. Please indicate hemagglutination “of strain QT598”.

L298. Please add “Fig 5A”.

L301. Please delete Fig S5

L303. Please delete "Fig 5A and S5" as these figures do not contain results of experiments with different concentrations of fucose and galactose.

L341-343. This should be moved to Material and Methods

L343. MT78 was not found in the list of strains (Table 1)

L365. QT5320 was not found in the list of strains (Table 1)

L373. “expressing PlfG classII” is already indicated L368.

L407. Black arrows can not be seen in Fig 7.

L479. Delete “also” .

L671. Prior to homogenization, was BSG or any other solution added to tissue samples ? (which volume ?)

L671. Cultures or tissues ?

L704. Please provide plasmid name and source.

L717. “washing”

Reviewer #2: 1. Some typos and minor language issues throughout (e.g., “exression”, “moeities”, “PlgG” instead of PlfG, occasional double words like “also also”).

2. Ensure consistent terminology: PL fimbriae, PLF vs plf (gene cluster), PlfG (protein).

3. In the Methods:

The description of plasmids, strains, and primers is very detailed (Tables 1–2, pp. 28–31). This is fine, but a brief schematic overview of the mutant/complement construction (e.g., in a figure) might help readers.

4. Check all abbreviations are defined at first use (e.g., MRHA, CFU/ml, LBA).

5. For Fig 4 and Fig 5, specify magnification/scale bars and number of independent experiments quantified.

6. For Fig 7, make clear that each glycan is spotted in quadruplicate and that binding was observed in 3/4 spots, etc.

7. Reference 10 requires correction, as the authors’ names are not included. Please provide the full citation details.

8. The Author Summary is essentially a duplication of the abstract. This section should be revised to communicate the significance of the findings in clear, non-technical language appropriate for the general public, rather than mirroring the scientific content of the abstract.

Reviewer #3: Title: It does not represent the content of the study well in my opinion. The authors compare adhesion to chicken and turkey lung frozen sections, so they define host range within avian hosts, but I’m not sure how the zoonotic potential is investigated. It might be that everything becomes clear, when authors rewrite the results section.

All figures- quality is too low. High resolution images are needed.

Line 39 Which tissue did you compared to show tissue-specificity of Plf class II?

Line 42 qRT-PCR - Please, check the MIQE guidelines and correct accordingly in the whole manuscript

Line 48-50 - The zoonotic potential is not clear for me.

Line 113-115 and 120-122 - Isn’t that the same information repeated twice?

Line 133- 144 - Remove this section as it contains M&M and isn’t relevant to the manuscript.

Line 145-162 - Remove this section (too much M&M) and mention the strain construction in 1-2 sentence in the next section.

Fig. 1 – merge with Fig. 2. Provide bull blot image as supplementary data.

Lien 197-199 – discussion. Is there possibility that this strain spreads through peritoneum not blood?

Fig. 2 – did authors considered consulting statistician and discuss potential outlier removal?

Line 230-232 – Why are these data not shown in the manuscript?

Line 234-236 – Shouldn’t you compare expression of this fimbriae in different tissues/gut to make such a statement? Right now you have proof that host tissue induces expression of Plf

Line 248 – Please include 1 sentence to introduce the reader into the reason why this experiment was performed

Line 251 - “Many bacteria”, but where are quantitative data for results from this section?

Fig. 4 - The use and organization of strains is not clear for me. Why not use wild type and Δplf in the analysis? Maybe also try to explain characteristics of the strains in figure caption? Please, add quantitative data for the results shown in these images.

Line 277 - “Lung tissue section from chicken, clone expressing Plf high-copy (QT6131).” A good example how clear figure caption is.

Line 286-289 - Where are these results shown?

Line 289-290 – I would add few words to explain what metaperiodate treatment does to erythrocyte glycans.

Line 291- Figure S3 and S4 are low quality

Lien 292-294 – why you decide to use Plf class 1 in this experiment. It wasn’t use in the animal work

Line 298-301 - Where are these results shown?

Line 302 – “HA” was that abbreviated earlier in the text?

Line 303-305 - Where are these results shown?

Line 312-314 – please, move to M&M.

Line 315-316 – Where is a barplot with results? By how many precent was the attachment reduced?

Line 335 – Please include 1 sentence to introduce the reader into the reason why this experiment was performed. Why use HEK293? Why use Plf class I and II?

Line 341- 343 - please, move to M&M.

Fig. 6 - What is MT78?

Line 358- 362 - please, move to M&M.

Line 364-367- I would move positive control to supplementary data. Also, reduce the text describing positive control in the text.

Fig. 7. Why not show the whole array in the figure?

Line 415 - 417 - please, move to M&M.

Line 417- 424 – Is that discussion or do you have results for this part?

Line 475 - 476- are these fimbriae plasmid or chromosome endoded?

Line 481- 493- How this paragraph discusses results from this manuscript?

Line 499 - Have you tested adherence to other types of tissues?

Line 501 – “local tissue colonisation” – Shouldn’t the mutant be then more systemically spread. I think that this is associated with the model selected for the study.

Line 511 - Citation for this statement would be helpful

Line 536 – Did you fin them in human E. coli isolates?

Line 548 – 553 –

Line 556-557 – On which figure have you shown these results?

Line 575 – citation 50, it that correct one?

Reviewer #4: Comment (minor)

Fig 3 and Fig 6. Data points for independent experimental replicates should be clearly shown as dots, rather than hidden by bars.

PLOS authors have the option to publish the peer review history of their article (what does this mean? ). If published, this will include your full peer review and any attached files.). If published, this will include your full peer review and any attached files.

**Do you want your identity to be public for this peer review?** For information about this choice, including consent withdrawal, please see our For information about this choice, including consent withdrawal, please see our Privacy Policy ..

Reviewer #1: No

Reviewer #2: **Yes:** Roxane Maria Fontes PiazzaRoxane Maria Fontes Piazza

Reviewer #3: No

Reviewer #4: **Yes:** Mark SchembriMark Schembri

**Figure resubmission:**
---

## [Decision Letter · Decision Letter 1]

24 Mar 2026

Dear Pr. Dozois,

We are pleased to inform you that your manuscript 'Host range and zoonotic potential linked to P-like fimbrial (PLF) adhesin specificity in avian pathogenic Escherichia coli' has been provisionally accepted for publication in PLOS Pathogens.

Best regards,

Eric Oswald, Ph.D., D.V.M.

Academic Editor

PLOS Pathogens

Thomas Guillard

Section Editor

PLOS Pathogens

Sumita Bhaduri-McIntosh

Editor-in-Chief

PLOS Pathogens

orcid.org/0000-0003-2946-9497

Michael Malim

Editor-in-Chief

PLOS Pathogens

orcid.org/0000-0002-7699-2064

Reviewer Comments (if any, and for reference):

Reviewer's Responses to Questions

**Part I - Summary**

Reviewer #1: The authors have answered the questions raised previously.

Reviewer #2: After analyzing the revised manuscript, I consider that the authors have adequately addressed the issues I raised. Therefore, the manuscript can be accepted for publication.

Reviewer #3: Summary doesn't change in comparison to previous stage

Reviewer #4: This revised manuscript addresses the minor comments raised in my first review. The authors also appear to have addressed the key questions raised by the other reviewers, and they have added additional revisions that improve the clarity of the manuscript. I commend the authors on an excellent study.

**Part II – Major Issues: Key Experiments Required for Acceptance**

Please use this section to detail the key new experiments or modifications of existing experiments that should be absolutely required to validate study conclusions.required to validate study conclusions.

Reviewer #1: No major issues.

Reviewer #2: No more issues

Reviewer #3: 1. Authors selectively respond to my comments. Not all my comments from the first stage were addressed.

2. Authors still do not provide quantitative data for their results.

3. Figure quality is low in several cases which makes it difficult for me to interpret the results.

4. It is unclear what experiments were performed/published previously, and which were performed for this study.

5. It is unclear what authors define as tissue-specific colonisation and I don’t think that the experimental design in this work allows to make such statements.

Authors can find detailed explanation of my major issues with all other issues below.

Reviewer #4: None

**Part III – Minor Issues: Editorial and Data Presentation Modifications**

Reviewer #1: No minor issues.

Reviewer #2: No more issues

Reviewer #3: Line 45 qRT-PCR - Please, check the MIQE guidelines and correct accordingly in the whole manuscript

Line 129 – why is figure S9 cited here? I do not understand how that can help to understand that adhesin proteins share only 56 % amino acid identity.

Line 141 -142 – I found in the cited publication mention of using LB broth. Was the test with LB agar shown in the cited previous study [19]?

Line 149 – in my understanding figures should help to visualize the differences observed and described by the authors. Here, we have statement that certain conditions increase expression of PLF, but the results are shown as 3 separate figures/subfigures. This could be improved.

Line 153-154 – Was the plf deletion mutant generated in previous work? From this text is seems like it was made for this study.

Fig. 1A. Is full blot shown as supplementary figure? Where is an WB image of the protein loading control to make?

Line 197-199 – As far as I understand, you have no statistical differences between WT and mutant in liver and spleen. Isn’t this an overstatement?

Line 209 – “tissue-specific colonization of the avian respiratory tract” – I would agree that there is tissue specific colonisation if authors would show any major differences between different compartments within respiratory tract or if authors would provide results regarding other tissues in other systems (gastrointestinal). I think that the results are interesting, but what they show is that contact with host tissue increases expression of PLF. Taking into consideration that authors inject the bacteria into respiratory tract they can observe the differences.

Line 238 - In their previous work they also shown increased expression of PLF in bladder of mice. Additionally, here authors only test airsacs and lung. For me the results indicate that host tissue (so far I don’t see much specificity to respiratory tract) induces expression of PLF

Line 251 – “tissue-specific adherence” – Maybe we understand tissue specificity differently. Please, explain me what you mean by “tissue-specific adherence”. I agree that you test here host specificity but not tissue specificity.

Line 264-267 – please show the results as barplot with statistics. Please move M&Ms description from the results to methods section.

Line 273-281, 329-334 - I’m not sure what that is. Did authors checked the pdf generated during submission, before final submission.

Line 288 - Is that strain QT4420?

Figure S3 and S4 – low quality images

Fig. 5A- Maybe it’s due to the quality of the figure, but I can’t see the difference. Is there any other way to present these results?

Fig. 5B - please show the results as barplot with statistics.

Figure 1, 2, 4 and 5 are low quality. Please improve the resolution.

Reviewer #4: None

PLOS authors have the option to publish the peer review history of their article (what does this mean? ). If published, this will include your full peer review and any attached files.). If published, this will include your full peer review and any attached files.

**Do you want your identity to be public for this peer review?** For information about this choice, including consent withdrawal, please see our For information about this choice, including consent withdrawal, please see our Privacy Policy ..

Reviewer #1: No

Reviewer #2: No

Reviewer #3: No

Reviewer #4: **Yes:** Mark SchembriMark Schembri

---

## [Editor Report · Acceptance letter]

Dear Pr. Dozois,

We are delighted to inform you that your manuscript, "Host range and zoonotic potential linked to P-like fimbrial (PLF) adhesin specificity in avian pathogenic Escherichia coli," has been formally accepted for publication in PLOS Pathogens.

Best regards,

Sumita Bhaduri-McIntosh

Editor-in-Chief

PLOS Pathogens

orcid.org/0000-0003-2946-9497

Michael Malim

Editor-in-Chief

PLOS Pathogens

orcid.org/0000-0002-7699-2064